

# Effect of extreme El Niño events on the precipitations of Ecuador

Dirk R. Thielen[1], Paolo Ramoni-Perazzi[2,3], Mary L. Puche[1], Marco Marquez[1], José I. Quintero[1], Wilmer Rojas[1], Alberto Quintero[4,5], Guillermo Bianchi[6], Irma A. Soto-Werschitz[7], Marco Aurelio Arizapana-Almonacid[8]

[1]Laboratory of Landscape Ecology and Climate, Venezuelan Institute for Scientific Research (IVIC), Caracas, 1020A, Venezuela
[2]Progetto SECOSUD II della Cooperazione Italiana (Sapienza - Eduardo Mondlane), Maputo, Mozambique
[3]Simulation and Modelling Center (CESIMO), University of Los Andes, Mérida, 5101, Venezuela
[4]Institute of Biodiversity, Conservation and Natural Resources Management, National Experimental University of Los Llanos "Ezequiel Zamora" (UNELLEZ), Barinas, Venezuela
[5]Center of Chemical Medicine, Venezuelan Institute for Scientific Research (IVIC), Caracas, 1020A, Venezuela
[6]Laboratory of Insect Ecology. Department of Biology, University of Los Andes, Mérida, 5101, Venezuela
[7]Departamento de Ecologia e Conservação, Instituto de Ciências Naturais, Universidade Federal de Lavras, Lavras, 37200-900, Minas Gerais, Brazil
[8]Research Group on Remote Sensing and Mountain Ecology, School of Engineering and Environmental Management, National Autonomous University of Huanta, Ayacucho, Peru

*Correspondence to*: Dirk R. Thielen (dirkthielen@gmail.com)

**Abstract.** Extreme El Niño events stand out not only for their powerful impacts but also because they are significantly different from other El Niños. In Ecuador, such events are accountable for impacting negatively the economy, infrastructure, and population. Spatial-temporal dynamics of precipitation anomalies from various types of extreme El Niño events are analyzed and compared. Results show that for Eastern Pacific and Coastal El Niño types, most precipitation extremes occur in the first half of the second year of the event. Any significant difference between events becomes more evident at this stage. Spatially, for any event, 50% of all extreme anomalies occurred at elevations <150m. Difference between events was significant when considering the altitude when reaching 80% of all extreme anomalies: EP-EN 97/98 at 500m, COA-EN 17 at 800m, and EN 82/83 at 1000m. Nevertheless, in some sectors of the Andean Cordillera, the ENSO signal could be detected at 3200-3900m. Distance to coastline and steepness of relief may play determining role. At lowlands, anomalies are most severe in regions where seasonality index is the highest. These results are useful at different decision-making levels for identifying most appropriate practices reducing vulnerability from a potential increase in extreme El Niño frequency and intensity.

## 1 Introduction

El Niño is the positive (warm) phase of ENSO (El Niño Southern Oscillation), characterized as a complex phenomenon of variable extent and intensity and contrasted impacts, from regional to global. It is originated by unusual warming of Sea Surface Temperature (SST) from the center of the equatorial Pacific Ocean to the coasts of Peru and Ecuador, bringing anomalously heavy rainfall to this region (**Gelati et al., 2014; Zambrano-Mera et al., 2018**), and are associated with substantial



socioeconomic impacts (**Cai et al., 2020**). Because of different locations of maximum SST anomalies and associated atmospheric heating, El Niño events are classified as eastern and central Pacific warming events (**Wang et al., 2016**). Eastern Pacific El Niños tend to have a stronger effect than Central Pacific El Niños, linked to the anomalous SST and convection (**Cai et al., 2020**). According to these authors, during an Eastern Pacific El Niño, anomalously warm waters (+ 1.5–3.0 °C) are often

observed adjacent to equatorial South American Pacific coast. Through transferring heat from the ocean to the atmosphere, this anomalous warming elevates air temperatures in the coastal region, triggering localized atmospheric convection and heavy rainfall.

But, such atmospheric–oceanic coupling at the South American Pacific coast is not exclusive to Eastern El Niño events. The Coastal is the least frequent type of El Niño event but has a proven capability of generating extreme precipitations in Ecuador

and Peru (e.g., the Coastal El Niño 2017). This type of extreme event is very rare and, until 2017, only two Coastal El Niños were previously reported in 1891 and 1925 (**Takahashi and Martínez, 2017**). While recent attention has been brought to the Central Pacific and the Eastern Pacific El Niño events, the Coastal El Niño represents another facet of ENSO that requires further study (**Takahashi et al., 2018a; Takahashi et al., 2018b**). The Coastal El Niño 2017 was an exceptional marine heatwave that did not last very long (only three months) but exhibited SST anomalies higher than any other extreme El Niño

event (+ 7°C) (**Pietri et al., 2021**). Differently from the Eastern Pacific El Niños, which are formed by the downwelling of equatorial Kelvin waves, Coastal El Niño 2017 seemed to have been formed by a marine heatwave generated by a local decrease of the winds in the Eastern Pacific, close to the coast of Ecuador and Peru (**Echevin et al., 2018; Hu et al., 2019**).

Before 1972, and more specifically to 1982, the 20th Century was dominated by rather mild El Niño events. Thus, El Niño was viewed as a regional phenomenon that interested mainly climate specialists. It was after learning from their devastating effects

from the occurrence of two of the most extreme El Niño events in history (i.e. 1982/83 and 1997/98 events) that the term El Niño, not only became familiar to the general public, but was seriously brought to the considerations of governments and policymakers worldwide (**Glantz, 2015; Hameed et al., 2018**).

Extreme El Niño events have a proven capacity, not only to severely affect the local climate, but to have an impact at a global scale through oceanic and atmospheric teleconnections (**Dewitte and Takahashi, 2019**). Extreme El Niño events stand out

not only for their powerful impacts, but also because they are significantly different from other El Niños (**Hong et al. 2014; Hameed et al., 2018**). Now, the main difficulty in the investigation of extreme El Niño dynamics has been related to the fact that, by definition, these occur rarely and very few have been observed comprehensively, and there is still not a clear picture of the extreme El Niño teleconnection complexity (**Dewitte and Takahashi, 2019**). Over the satellite era, only three strong El Niño events (excluding the Coastal type) have been observed (1982/83, 1997/98, and 2015/16). Nevertheless, and despite their

limited number, extreme El Niño events may share some common features (**Hong, 2016**). Therefore, the group study of extreme El Niño provides an excellent starting point to understand the complexity of El Niño phenomenon.

With a current population of around 18 million and a continental surface of 248,540 km², Ecuador is located between 1.5°N and -5.0°S, comprising an important extension of both, the tropical South American Pacific coast and the Amazon basin. These two geographical regions are separated by the Andean Cordillera which crosses the country from north to south and constitutes



a substantial topographic barrier (**Morán-Tejeda et al., 2016**). Regarding the presence of the Andean mountain chain, this most certainly modifies the ENSO signal (**Vuille et al., 2000; Morán-Tejeda et al., 2016; Tobar and Wyseure, 2017; Quishpe-Vásquez et al., 2019**). However, there is no clear explanation of how far into the Andes the effects of the ENSO are perceived (**Morán-Tejeda et al., 2016**). Detailed basin-wide assessments of the influence of ENSO in the transition from the

5 coastal plain toward the western Andean Cordillera are also very scarce (**Pineda et al. 2013**). Various studies have considered precipitation in Ecuador (**Rossel et al., 1999; Bendix and Bendix, 2006; Buytaert et al., 2006**), or have focused on precipitation in specific areas of the country (**Mora and Willems, 2012; Thielen et al., 2015 and 2016**). In general, a strong connection between the ENSO and precipitation in Ecuador has been found, but none of the previous studies analyzed stations throughout the entire country (**Morán-Tejeda et al., 2016**). According to these authors, a study of trends and variability in

precipitation, based on up-to-date data for the entire country, is still lacking. A situation that is especially true when considering the different effects on spatial and temporal precipitation dynamics resulting from the occurrence of various types of extreme El Niño events (**Thielen et al., 2021a**).

In Ecuador, historical extreme El Niño events are accountable for generating very important, direct or indirect, negative effects on the economy, infrastructure and the population (**OPS-OMS, 1999**). Most economic costs are related to losses of agricultural

production and damages to infrastructure (US$ 640.6 million in 1982/83, and US$ 2,882 million in 1997/98). Around 60% of the population of Ecuador may have their living conditions altered, directly or indirectly, by the occurrence of an extreme El Niño event. They have been responsible for generating mayor migratory waves (**CEPAL, 1998**). For instance, over one million Ecuadorians fled the country after affectation on Ecuador´s economy due to El Niño in 72/73 and 82/83 (**Bernabé et al., 2014**), as well as in 97/98 (**OPS, 2000**). Extreme El Niño events in Ecuador have also been accountable for important epidemics of

diverse vector-transmitted and infectious diseases such as cholera, leptospirosis, dengue, chikungunya, zika, malaria, etc. (**Gabastou et al., 2002; The World Bank Group, 2011**).

Now, in response to greenhouse warming, extreme El Niño events are projected to double their occurrence, while a less pronounced increase is projected for moderate El Niño events (**Gulizia and Pirotte, 2022; Cai et al., 2014**). Likewise, climate model projections also indicate an increase in the frequency of extreme Coastal El Niño (**Peng et al., 2019**). Of course, these

25 impacts on climate extremes as well as the associated socioeconomic impacts would also take place much more frequently too (**Gulizia and Pirotte, 2022; Lopez et al., 2022**).

Taking into account the implications for the Ecuador of such a forecast, the main objective of the present study was to analyze and compare, based on up-to-date data for the entire insular and continental territory, the dynamics of precipitation anomalies resulting from the various types of extreme El Niño events, including the Coastal El Niño. The results were discussed regarding

the spatial and temporal variability generated by the topographic gradients of the dorsal of the Andes at both, the Pacific slope and the Amazon slope, as well as in terms of their principal basins or hydrological systems. The results provide solid and opportune evidence that can be used at different decision-making levels for identifying, in the context of global climate change scenarios, the most appropriate practices for reducing vulnerability and risks from a potential increase in extreme El Niño frequency and intensity.



## 2 Material and Methods

### 2.1 Study area

The study area was defined as the totality of the continental and insular (offshore) territory of Ecuador. As for the continental territory, this was first divided into two main and distinctive zones: The *Pacific slope* (116,592 km$^2$) and the *Amazon slope*
(131,948 km$^2$). Delineation of these zones was defined by the dorsal of the Andes, a dominant orographic barrier determining if runoff from rainfall is to be drained to the Pacific shore or the Amazon basin. Following **CNRH (2002)** classification system, each of these two continental zones was further divided into hydrographic systems or basins regarding their climate and spatial homogeneity. Through GIS applications, data from HydroSHEDS (http://hydrosheds.cr.usgs.gov) was used to delineate each basin. This resulted in 23 hydrographic systems for the *Pacific slope*, and 7 for the *Amazon slope* (see **Fig. 1**). Finally, regarding
the insular territory, a unique hydrographic system was established encompassing all offshore islands, specifically the *Galapagos* Islands (8,233 km$^2$).

### 2.2 Data

Precipitation data was obtained from the Climate Hazards Group Infrared Precipitation with Stations (CHIRPS V2.0, https://iridl.ldeo.columbia.edu/SOURCES/.UCSB/.CHIRPS/.v2p0/.monthly/.global/). CHIRPS V2.0 is a quasi-global gridded
rainfall time series dataset, spanning 50°S-50°N, from 1981 to near-present, 0.05° resolution satellite imagery with *in situ* station data, with great applications in monitoring precipitation extremes **(Funk et al., 2015)**. According to **Beck et al. (2017)**, in a global-scale evaluation of 23 precipitation datasets, CHIRPS V2.0 tends to perform the best in hydrological modeling of tropical regions, specifically in Central and South America. As for Ecuador, **Thielen et al. (2021a)** successfully tested its applicability in the spatial/temporal analysis of hydroclimatological extreme events in one of the most important and extended
basins of the Ecuadorian Pacific slope. For the present study, monthly data for the time series Jan-1981/Dec-2018 were obtained from 456 rasters. Monthly and annual mean, as well as some other basic precipitation parameters, were obtained through GIS applications.

### 2.3 Calculation of the Standardized Pluviometric Drought Index - SPDI

In this study, the precipitation spatial-temporal dynamics was analyzed by the Standardized Pluviometric Drought Index
(SPDI) developed by **Pita (2001)**. The SPDI is a monthly rainfall index that is based on the calculation of cumulative monthly rainfall anomalies, similar to the well-known Standardized Precipitation Index (SPI) of **McKee et al. (1993)**, more specifically, the 12-month SPI. As in this index, values ranging from +1 to +1.5 and +1.5 to +2.0 are associated with moderately humid and very humid episodes, respectively, and values exceeding +2 are representative of extremely humid episodes. Moderately dry, very dry, and extremely dry spells are characterized by the same ranges with a negative sign (see **Table 1**). The SPDI is
calculated as follows:

First stage, Eq. (1):



$$APi = Pi - P_{MED} \tag{1}$$

Where $APi$ is the monthly precipitation anomaly, $Pi$ is the monthly precipitation, and $P_{MED}$ is the median precipitation of the month for the series. As for this study: 1981-2010.

Second stage, Eq. (2):

$$APAi = \sum APi \qquad \text{From } i = \text{negative } AP \text{ to } i = \text{positive } AP \tag{2}$$

Where $APAi$ is the accumulated precipitation anomaly of the month.

Third Stage, Eq. (3):

$$SPDI = (APAi - \overline{APA})/\sigma APA \tag{3}$$

Where $\overline{APA}$ is the average value of accumulated precipitation anomalies of all the months of the series, and $\sigma APA$ is the standard deviation of accumulated precipitation anomalies of all the months of the series.

GIS applications allowed us to implement these equations to the 456 aforementioned CHIRPS V2.0 rasters and generate SPDI products such as images of 0.05° resolution or monthly zonal values, and at different space and/or time criteria. In the present study, monthly values of SPDI were estimated, based on the 1981-2010 climatology, for the three main zones: *Pacific slope*, *Galapagos*, and *Amazon slope*; as well as for each of the 30 continental hydrological systems (see **Fig. 1**). Analysis of monthly SPDI dynamics was performed in the two-year series comprising each extreme El Niño event. The significance of the statistical difference between monthly precipitation and/or SPDI values for any pair of extreme events, in each of the three zones, was identified by two-tailed Paired *t*-Tests and an $\alpha = 0.05$. While similarities in spatial-temporal dynamics of SPDI between the 30 hydrological systems were identified through Cluster Analysis (K-means clustering using Euclidean distance). **Gong and Richman (1995)** noted that nonhierarchical methods, such as the K-means algorithm, outperformed hierarchical methods (Ward's and the average linkage methods) when tested with precipitation data, as well as for SPI series **(Santos et al., 2010)**.

**2.4 Altitudinal dynamics of SPDI**

As for continental Ecuador and through geoprocessing tools available from GIS software, results from SPDI estimations for each extreme El Niño event were combined with rasterized altitude data obtained from SRTM 1 Arc-Second Global (approx. 30m resolution, and freely available at https://earthexplorer.usgs.gov/), and then resampled at CHIRPS resolution (*ie*. 0.05°). The frequency of pixels with SPDI ≥2.0 was determined along the entire altitudinal gradient, for both the *Pacific* and the *Amazon slope*, and for each extreme El Niño event. The significance of the statistical differences between any pair of such SPDI spatial dynamics was identified by two-tailed Paired t-Tests and an $\alpha = 0.05$.

**2.5 Seasonality Index ($\overline{SI}$)**

The seasonality of precipitation in continental Ecuador was estimated by the Seasonality Index ($\overline{SI}$) **(Walsh and Lawler, 1981)** which quantifies the variability in monthly precipitation throughout the year. It is estimated by the sum of the absolute deviations of mean monthly precipitations from the overall monthly mean, divided by the mean annual precipitation, Eq. (4):





$$\overline{SI} = \frac{1}{\bar{R}} \sum_{N=1}^{n=12} |\bar{x}_n - \bar{R}/12| \tag{4}$$

Where $\bar{x}_n$ is the mean precipitation of month $n$ and $\bar{R}$ is the mean annual rainfall. The $\overline{SI}$ can vary from zero (if all months have equal precipitation amounts) to 1.83 (if all the rainfall occurs in a single month). Thus, the higher the $\overline{SI}$, the more seasonal or concentrated the precipitations are. The relationship between resulting seasonality index and the anomalies of precipitation

(as SPDI) was evaluated at basin level.

**2.6 Definitions of extreme El Niño events: The mega-El Niño and the Coastal El Niño**

The Oceanic Niño Index (ONI) is NOAA's primary indicator for monitoring ENSO. It is based on the monitoring of sea surface temperatures (SSTs) in the central Pacific Ocean and is used to identify the onset of an above-average SST threshold that persists for several months, encompassing both the beginning and end of an El Niño episode (**Glantz and Ramirez, 2020**).

The ONI tracks the running 3-month average sea surface temperatures (SST) in the east-central tropical Pacific (120°–170°W, 5°S–5°N), specifically the NIÑO 3.4 region. El Niño occurs when the anomalies exceed +0.5°C for at least five consecutive months. The threshold is further broken down into Weak (with a +0.5 to +0.9 SST anomaly), Moderate (+1.0 to +1.4), Strong (+1.5 to +1.9), and Very Strong (≥ 2.0) events. As for Very Strong or mega-El Niño events, the SST anomalies may be +2.0°C for several months (**Chen et al., 2017**). According to these parameters, three mega-El Niño events were identified since 1951:

1982/83, with mean ONI from October 1982 to February 1983 of 2.09°C; 1997/98, with mean ONI from September 1997 to February 1998 of 2.20°C; and 2015/16, with mean ONI from October 2015 to February 2016 of 2.22°C) (data source http://www.cpc.ncep.noaa.gov/data/indices/oni.ascii.txt). Additionally, regarding where the SST anomalies peak, El Niño events can be classified as eastern Pacific (EP), involving the easterly NIÑO 1+2 region, and the central Pacific (CP), mainly NIÑO 3.4 (**Larkin and Harrison, 2005**; **Ashok et al., 2007**; **Kug et al., 2009**). Although all aforementioned mega-El Niño

events have been EP El Niño events, the 2015/16 event should be considered as a mixed regime of both EP and central Pacific (CP) El Niño (**Santoso et al., 2017; L'heureux et al., 2017; Vicente-Serrano et al., 2017; Wang et al., 2020**), mainly due to an erratic response of SST anomalies peaking of SST (**Xie and Fang, 2019**). As for the present study, the three mega-El Niño events are referred to as follows: **EP-EN 82/83,** for El Niño 1982/83; **EP-EN 97/98,** for El Niño 1997/98; and **MIX-EN 15/16**, for El Niño 2015/16.

Besides tropical Pacific El Niño events (EP or CP), the study area is also affected by a more local type of El Niño event: The Coastal El Niño, a very rare and unique event which develops differently from either CP or EP El Niño events. To distinguish the Coastal El Niño from the warm ENSO phase, **ENFEN (2012)** operationally defines the Coastal El Niño based on the seasonal NIÑO 1+2 SST anomaly: 3-month running-mean Niño-1+2 SST above 0.4°C for three or more consecutive months. A strong Coastal El Niño developed off the coast of Peru from January to April 2017 (**ENFEN, 2017**; **WMO, 2017a,b**;

**Takahashi and Martínez, 2017**; **Ramírez and Briones, 2017**; **Garreaud, 2018**) (hereafter **COA-EN 17**), and has been the strongest on record, and developed rather fast and unexpectedly from the warming of SST specific to far eastern tropical Pacific.



### 3 Results

#### 3.1 *Pacific slope*

#### 3.1.1 Precipitation

Historical mean annual precipitation for the *Pacific slope* resulted in 1348mm (series 1981-2010). About 70% of annual
precipitations occurs during the first four months of the year (Jan-Apr). According to **Walsh and Lawler (1981)**, precipitations
here are markedly seasonal with a long drier season ($\overline{SI}$ = 0.96). As appreciated in **Fig. 2-Ia**, mean monthly precipitation during
the first year (Year 1) of any extreme El Niño event does not differ significantly from that of historical mean (P<0.05). It is
during the second year (Year 2) of El Niño event, specifically the first half, when precipitations significantly differ from that
of historical values. For example, annual precipitation for Year 2 of mega-El Niño events **EP-EN 82/83** (*ie*. 1983) and **EP-EN**
**97/98** (*ie*. 1998) resulted significantly higher (2483mm, *P*=0.003; and 2590mm, *P*=0.022; respectively) than the historical
mean. Although precipitation values between mega-El Niño events **EP-EN 82/83** and **EP-EN 97/98** were not significantly
different (*P*=0.194), values for Year 1 was significantly drier in **EP-EN 82/83** than in **EP-EN 97/98** (1234 vs. 1609mm,
*P*=0.042). On the other hand, during the other mega-El Niño event, **MIX-EN 15/16**, neither Year 1 nor Year 2 presented
annual precipitations significantly different than those of the historical mean (1368mm, *P*=0.868; and 1299mm, *P*=0.666;
respectively). As for the Coastal El Niño, **COA-EN 17**, annual precipitation for the year 2017 tended to be significantly
different from historical mean (2072mm, *P*=0.097).

#### 3.1.2 SPDI

The resulting SPDI temporal dynamics between the different extreme El Niño events are given in **Fig. 2-Ib**. As a result of
precipitation dynamics during Year 1 (**Fig. 2-Ia**), none of the mega-El Niño events and neither the Coastal El Niño showed
SPDI values different than the "near normal" condition (-0.99 – 0.99). On the other hand, the *Pacific slope* experienced
"extremely humid" (SPDI ≥2.00) during Year 2 of **EP-EN 82/83** and **EP-EN 97/98**, with mean SPDI values not significantly
different: 2.02 and 2.19, respectively (*P*=0.431). As for **EP-EN 82/83**, precipitation anomalies started in March 1983 (SPDI
=1.64) and lasted for ten months until December of that year (SPDI =1.12). For seven consecutive months (Apr-Oct), **EP-EN
82/83** presented a sustained extremely humid condition (SPDI ≥2.00). In 1983, 54.9% of continental Ecuador was affected by
this extreme precipitation anomaly, from which, 93.8% comprised the *Pacific slope* (**Fig. 3-I**). In this easterly slope, about
90% of such extreme precipitations occurred at altitudes of 1900 m or less, and over half at less than 200 m asl (**Fig. 4**).
Even though precipitation anomalies during **EP-EN 97/98** also lasted ten months (Feb-Nov/1998), extreme humid conditions
(SPDI ≥2.00) for this mega-El Niño was reduced to six months (Mar-Aug), reaching a maximum SPDI value of 3.56 by June
1998 (**Fig. 2-Ib**). It is also evident from this figure that extremely wet SPDI values were obtained for the *Pacific slope* earlier
30  in **EP-EN 97/98** than in **EP-EN 82/83** mega-El Niño. About 75.6% of continental Ecuador was affected by precipitation
anomalies of SPDI ≥2.00 during Year 2 of **EP-EN 97/98**. As in previous mega-Niño, the most comprised area was the *Pacific*





*slope* (98.8% of total area, **Fig. 3-II**). On this slope, during 1998, 90% of extreme precipitations occurred at altitudes of 1300m or lower, and about 50% of such precipitations, at coastal areas with elevations less than 150masl (**Fig. 4**).

Precipitation during mega-El Niño **MIX-EN 15/16** resulted in SPDI dynamics for Year 2 (*ie.* 2016) being extremely different ($P<0.0001$) from Year 1 and Year 2 of both, **EP-EN 82/83** and **EP-EN 97/98**. During this mixed mega-El Niño, the SPDI

value for 2015 tended to be similar to that of 2016 (0.41 and 0.33, respectively; $P=0.051$), for a resulting "near normal" precipitation condition for the entire duration of this extreme El Niño event (**Fig. 2-Ib**).

During 2017, Coastal El Niño **COA-EN 17** generated a precipitation anomaly lasting five months (Mar-Jul), with an SPDI maximum of 2.23 for about two months (**Fig. 2-Ib**). SPDI dynamics during 2017 was significantly different from that of Year 2 of **EP-EN 82/83** and **EP-EN 97/98** ($P<0.05$), affecting only 5.7% of continental Ecuador with extreme precipitation

anomalies of SPDI ≥2.00. Still, as in the two mega-El Niño EP events, over 88.7% of this area comprised the *Pacific slope* (**Fig. 3-III**). Extreme precipitation anomalies in **COA-EN 17** reached the breakpoint of 90% at an altitude of about the same as **EP-EN 82/83**, 1700masl. As in **EP-EN 97/98**, over 50% of all extreme anomalies occurred at elevations under 150 m (**Fig. 4**).

### 3.2 *Galapagos*

### 3.2.1 Precipitation

Historical annual mean precipitation for *Galapagos* resulted in 89mm (series 1981-2010), much drier than that of the *Pacific slope* (**Fig. 2-IIa**). About 82% of annual precipitation occurs from February to April, reflecting an extreme seasonality ($\overline{SI}=$ 1.28, **Walsh and Lawler, 1981**). As in the *Pacific slope*, precipitations in *Galapagos* during Year 1, for any of the extreme El Niño events considered, did not differ significantly ($P>0.05$) from that of the historical mean. On the other hand, during Year

2 of mega-El Niño events **EP-EN 82/83** and **EP-EN 97/98** (1983 and 1998, respectively), rainfalls about tripled that of the monthly mean value for the 30-years series 1981-2010 (267 and 284mm, respectively), an increase that tends to be significantly higher than the historical mean ($P≈0.06$). Annual precipitations among mega-El Niño events **EP-EN 82/83** and **EP-EN 97/98**, as for Year 1 and Year 2, did not differ significantly ($P=0.1390$ and 0.616, respectively). As for **MIX-EN 15/16**, this mega-El Niño event did not generate precipitations significantly different from those of the historical mean, neither in 2015 (96mm,

$P=0.260$) nor in 2016 (74mm, $P=0.641$). Likewise, the Coastal El Niño of 2017 (**COA-EN 17**) did not generate precipitations any different from that of the historical mean (131mm, $P=0.205$) for the *Galapagos* Islands.

### 3.2.2 SPDI

As for Year 1 in *Galapagos*, all of the mega-El Niño events generated an SPDI value "near normal" (-0.99 – 0.99). A situation that changed very significantly in Year 2 of both, **EP-EN 82/83** and **EP-EN 97/98**, when the mean condition turned to

30 "extremely humid": 2.13 in 1983, and 3.64 in 1998 (see **Fig. 2-IIb**). From this Figure, as well as from **Fig. 3-I** and **II**, it is evident for Year 2 an extremely significant difference ($P<0.0001$) in the temporal dynamics of SPDI values between **EP-EN**





**82/83** and **EP-EN 97/98** mega-El Niño events. For instance, in 1983, the "extremely humid" condition (SPDI ≥2.0) was reached abruptly in March and lasted until August that year when reached a maximum SPDI value of 3.46. From April to August 1983, 99% of the *Galapagos* was affected by an SPDI mean value of 3.33. As for 1998, overall affectation by excessive rainfall lasted 12 months. It started in February when precipitation generated a "very humid" condition (SPDI =1.64), and for

the next 11 months persisted an "extremely humid" condition with SPDI values ranging from 3.01 to 4.31, and affecting more than 98% of the *Galapagos* surface. Mega-El Niño **EP-EN 97/98** affectation lasted until the first months of 1999, after a sudden drop in SPDI values: from 3.82 in January to 0.06 in February. No significant effects on SPDI value dynamics were associated with the mega-El Niño **MIX-EN 15/16** event. As for the Coastal El Niño 2017, from April to June, precipitations generated an SPDI value reaching 1.01 to 1.03, a value that barely denotes a moderately wet condition (see **Fig. 2-IIb** and **Fig.**

**3-III**). SPDI monthly values dynamics of 1983 in the *Galapagos* was not significantly different from that in the *Pacific slope* (*P*=0.606) (see **Fig. 2-Ib** and **IIb**). On the other hand, in 1998, a very significant difference (*P*=0.003) in SPDI dynamics was observed between these two contrasting geographical regions. Such a significant difference was also observed in the resulting SPDI from Coastal El Niño 2017 between the *Galapagos* and the continental Ecuador.

### 3.3 *Amazon slope*

**3.3.1 Precipitation**

Historical annual mean precipitation (series 1981-2010) for the *Amazon slope* was 2824mm, more than double that of the *Pacific slope* (1348mm). From March to July mean monthly precipitation is around 10%, while for the rest of the year it is about 7% (see **Fig. 2-IIIa**). This results in a Seasonality Index of 0.26 which, according to **Walsh and Lawler (1981)**, is referred to places where precipitation spread throughout the year, but with a definite wetter season. As for precipitation in Year

1, both **EP-EN 97/98** and **MIX-EN 15/16**, showed values significantly drier (*P*<0.05) than the historical mean (2538 and 2269mm, respectively). While for Year 2, none of the extreme El Niño events, **EP-EN 82/83**, **EP-EN 97/98**, **MIX-EN 15/16**, or **COA-EN 17**, showed a significate value different from the historical (*P*>0.05). **MIX-EN 15/16** was significantly drier (2519mm, *P*<0.05) than that of **EP-EN 82/83**, **EP-EN 97/98,** and **COA-EN 17** (2978, 2801mm, and 3134mm, respectively).

**3.3.2 SPDI**

None of the three mega-El Niño events, nor the Coastal El Niño of 2017 generated SPDI values different than "near normal" (-0.99 – 0.99, **Fig. 2-IIIb**) in the *Amazon slope*. Mean SPDI values for Year 2 of **EP-EN 97/98**, **MIX-EN 15/16,** and **COA-EN 17** were significantly drier (-0.64, -0.69, and -0.19, respectively; *P*<0.0001) than **EP-EN 82/83** (0.89). No significant difference (*P*>0.05) was observed between SPDI values of Year 1 and Year 2 of either **EP-EN 82/83** (0.92 and 0.89) or **EP-EN 97/98** (-0.64 and -0.64). On the other hand, a very high significant difference was detected between SPDI values for the

30  years 2015 and 2016 (0.33 and -0.69, respectively; *P*<0.0001). SPDI for Year 2 of **EP-EN 82/83** was significantly higher than those of **EP-EN 97/98**, **MIX-EN 15/16,** and **COA-EN 17** (*P*<0.0001, see **Fig. 2-IIIb**). As for areas of continental Ecuador





with extreme precipitation anomalies of SPDI ≥2.00, around 6.2% occurred in the *Amazon slope* during Year 2 of **EP-EN 82/83,** 1.2% during **EP-EN 97/98,** and 11.3% during **COA-EN 17 (Fig. 3-I, II** and **III)**. No such extreme events were observed during **MIX-EN 15/16** for this slope during 2016. About 90% of all extreme precipitation during **EP-EN 97/98** occurred at altitudes rather high (2500-4000masl), while for **COA-EN 17** such extreme events encompassed a much wider altitudinal

gradient (1000-4000masl) (**Fig. 4**). Extreme precipitation anomalies were most spatially restricted during **EP-EN 82/83** for the *Amazon slope*: 64% comprised altitudes between 300 and 400masl and the remaining 39% from elevations >2500m. In the lowlands of the *Amazon slope*, the presence of precipitation anomalies observed in **Figure 3** (positive as in **EP-EN 82/83,** or negative as in **EP-EN 97/98** and **COA-EN 17**) was preexistent long before the beginning of any of these extreme El Niño events, thus unrelated to their dynamics (**CAF, 2000**).

**3.4 *Hydrological systems***

**3.4.1 EP-EN 82/83**

From cluster analysis of 1983 (i.e. Year 2) of this mega-Niño´s SPDI monthly data, four distinctive groups of hydrological systems are evident (**Table 2**). A first group conformed by 11 basins (Cluster 1: 16-NARANJAL PAGUA, 23-ISLA PUNA, 17-JUBONES, 14-TAURA, 13-GUAYAS, 15-CANAR, 12-ZAPOTAL, 21-PUYANGO, 18-SANTA ROSA, 19-

ARENILLAS, and 20-ZARUMILLA), all belonging to the *Pacific slope*, having an extremely high collective monthly mean SPDI from March to October of 2.89 (**Table 2-a**), with a mean affection of 89.0% of the area during that same time (**Table 2-b**). A second group of five *Pacific slope* basins (Cluster 2: 11-JIPIJAPA, 09-CHONE, 10-PORTOVIEJO, 07-MUISNE, and 08-JAMA) also showed an eight consecutive Mar/Oct pulse of an extremely high collective SPDI mean (2.47) (**Table 2-a**). But, differently than Cluster 1, in Cluster 2 the mean SPDI of the three first months of the Mar/Oct pulse was significantly

lower (*P*<0.05). As a result, basins from Cluster 2 have less extended affection areas during the trimester Mar/May (69.2%) (**Table 2-b**). Another difference in the SPDI response during mega-El Niño **EP-EN 82/83** among these two clusters of basins is the higher values reached in February in Cluster 1 compared to Cluster 2. No significant difference (P>0.05) was observed between these two clusters during the remaining months of 1983 (Jan, Nov-Dec).

Two additional clusters of hydrological systems resulted from the spatiotemporal analysis of 1983 SPDI monthly data. Cluster

3 is a group of three southerly *Amazon slope* basins (29-MORONA, 28-PASTAZA, and 30-SANTIAGO), where none of them showed significant precipitation anomalies (*i.e.* SPDI>1.0) (**Table 2-a**). Cluster 4, on the other side, showed SPDI dynamics that generated moderately to very humid conditions throughout the entire year, and were not confined only to the March/October pulse observed for Clusters 1 and 2. Cluster 4 comprises eleven basins from both slopes: four from the *Amazon slope* (31-CHINCHIPE, 26-NAPO, 27-CUNAMBO, and 25 SM PUTUMAYO); six from the northernmost section of the

*Pacific slope* (01-CARCHI, 02-MIRA, 05-VERDE, 03-MATAJE, 04-CAYAPA and 06-ESMERALDAS), and one to the southernmost basin of the *Pacific slope* (22-CHIRA, see **Table 2-a**). In this cluster, the most extreme precipitation anomalies





as well as a most extended area of affection occurred from Jun until Oct, and even until the end of 1983. This was mainly due to the SPDI dynamics of the *Pacific slope* basins (**Table 2-b**).

### 3.4.3 MIX-EN 15/16

As for the effects of mega-El Niño event **MIX-EN 15/16** on the hydrological systems´ precipitations, overall monthly SPDI

values for 2016 were well defined as near normal (-0.99 – 0.99). During this event, only five systems (08-JAMA, 09-CHONE, 19-ARENILLAS, 21-PUYANGO, and 20-ZARUMILLA) showed a short lasting (2 months, Mar/Apr) and discrete increase of SPDI values, barely reaching a moderately humid condition, with a collective SPDI mean of 1.12. While the *Pacific slope´s* systems tended to have positive SPDI values during 2016, the *Amazon slope´s* systems tended rather negative ones. As for 31-CHINCHIPE, this southernmost Amazon basin showed an overall 2006 SPDI value of -1.40, a moderately dry condition.

### 3.4.2 EP-EN 97/98

During 1998, that is Year 2 of **EP-EN 97/98** mega-El Niño event, two distinctive groups of hydrological systems showed prolonged and extremely high precipitation anomalies (**Table 3**). A first large group, conformed by fourteen *Pacific slope* basins (Cluster 1: 14-JUBONES, 06-ESMERALDAS, 21-PUYANGO, 08-JAMA, 07-MUISNE, 13-GUAYAS, 16-NARANJAL PAGUA, 14-TAURA, 15-CANAR, 12-ZAPOTAL, 19-ARENILLAS, 18-SANTA ROSA, 20-ZARUMILLA,

and 23-ISLA PUNA), with a Jan/Jul mean monthly SPDI of 3.68, was affected by a sustained extremely humid condition for six to seven consecutive months (Feb-Jul/Aug). For some basins of this first cluster, the affections of such extremely wet condition prolonged up to Oct (**Table 3-a**). Spatially, besides the northern basin of 06-ESMERALDAS (with only 39.7%), the rest of the basins of this first cluster were affected by extreme precipitation anomalies in about 92.7% of their areas (**Table 3-b**). A second smaller group of *Pacific slope* basins (Cluster 2: 10-PORTOVIEJO, 09-CHONE and 11-JIPIJAPA), presented a

Jan/Jul mean monthly SPDI condition as extremely humid as Cluster 1 (3.62 vs. 3.68). But, differently than Cluster 1, this group of three basins steadily prolonged their extremely humid condition until Nov, for a total span of ten months, resulting in a collective SPDI mean value significantly higher than for any cluster and for any other extreme El Niño event (**Table 3-a**), steadily affecting, fairly homogeneously, about 93% of the basins areas (**Table 3-b**).

On the other hand, the mega-El Niño event **EP-EN 97/98** did not appear to have generated humid precipitation anomalies for

the rest of continental Ecuadorian basins. Cluster 3, for example, during 1998 a nine-basin group from both *Amazon* and *Pacific slope* (03-MATAJE, 02-MIRA, 29-MORONA, 28-PASTAZA, 27-CUNAMBO, 30-SANTIAGO, 04-CAYAPA, 05-VERDE, and 22-CHIRA), showed a collective sustained SPDI mean value about normal (>-1, <1; **Table 3-a**). Moreover, during this mega-El Niño event, Cluster 4, a four-basins group (26-NAPO, 25-SM PUTUMAYO, 31-CHINCHIPE, and 01-CARCHI), showed a moderately dry mean precipitation condition (SPDI =-1.23, **Table 3-a**).



### 3.4.4 COA-EN 17

During the Coastal El Niño of 2017, a group of 15 *Pacific slope* basins (Cluster 1: 20-ZARUMILLA, 14-TAURA, 09-CHONE, 08-JAMA, 16-NARANJAL PAGUA, 15-CANAR, 13-GUAYAS, 17-JUBONES, 11-JIPIJAPA, 10-PORTOVIEJO, 12-ZAPOTAL, 23-ISLA PUNA, 22-CHIRA, 21-PUYANGO, and 18-SANTA ROSA) showed an extremely humid condition

from Mar to Jun (mean SPDI =2.63) (**Table 4-a**). During these four months, the Coastal El Niño event affected 78.4% of the area of the basins of Cluster 1. By July, the affected area was 47.0%, and then lowered to 13.3% for the rest of the year (**Table 4-b**). Precipitation anomalies of varying intensities extended until Sep for some of these basins of Cluster 1. In the case of 18-SANTA ROSA, the anomalies lasted until Dec.

From cluster analysis of SPDI dynamics during **COA-EN 17**, two other groups of basins from both *Pacific* and *Amazon* slopes

were identified - Cluster 2: 05-VERDE, 04-CAYAPAS, 03-MATAJE, 06-ESMERALDAS, 07-MUISNE, 25-SM PUTUMAYO, 02-MIRA, 27-CUNAMBO, 01-CARCHI, and 30-SANTIAGO; and Cluster 3: 29-MORONA, 28-PASTAZA, 26-NAPO, and 31-CHINCHIPE (**Table 4-a**). While Cluster 2 mean SPDI tended to be positive, and Cluster 3 negative, none showed precipitation dynamics that resulted, spatially and temporarily, in a mean condition different than normal. The last cluster, conformed exclusively by a rather small and southernmost *Pacific slope* basins (Cluster 4: 19-ARENILLAS), showed

a unique SPDI dynamic when, from Mar to Dec, reached a sustained extremely humid condition (mean SPDI =2.8), spatially affecting 98.9% of basin´s area from Mar to Jun, and 66.2% from Jul to Dec (**Table 4-a and b**).

### 4 Discussion

In this study, the application of the SPDI was most appropriate when analyzing and comparing temporal and spatial dynamics of precipitation extremes among different extreme El Niño events. Likewise, CHIRPS V2.0 was confirmed to be a valuable

source of monthly precipitation data for monitoring extreme events and contributed to the understanding of the spatial and temporal variability of monthly rainfall in Ecuador, as demonstrated for other extended South American regions (**Paredes-Trejo et al., 2016**; **Baez-Villanueva et al., 2018**; **Rivera et al., 2019**; **Thielen et al., 2020 and 2021b**).

For any of the considered regions (*Pacific slope*, *Galapagos*, and *Amazon slope*), and for any of the extreme El Niño events **EP-EN 82/83**, **EP-EN 97/98**, **MIX-EN 15/16,** and **COA-EN 17**, precipitations during the first year (Year 1: 1982, 1997, 2015

and 2016, correspondingly) was not significantly different from that of the historical annual mean (series 1981-2010) (**Fig. 2**). On the other hand, for both, *Pacific slope* and *Galapagos*, it is during the second year of **EP-EN 82/83**, **EP-EN 97/98** and **COA-EN 17**) (Year 2: 1983, 1998 and 2017, correspondingly), and more specifically during the first half of these years, coincidentally encompassing the rainy season, when most precipitation extremes occur. According to **Cai et al. (2020)**, the rainfall impacts on the coast of Ecuador and Peru occur mainly in the rainy months of February, March, and April, when

regional SSTs are seasonally at their highest, and the threshold for deep convection is more likely to be reached. As for Year 2 of **MIX-EN 15/16** (2016), there was no evidence of any significant precipitation anomaly generated on Ecuadorian territory by the occurrences of the mixed (EP-CP) type of extreme El Niño.





Regarding overall SPDI dynamics (series 1981-2020) at both, the *Pacific slope* and the *Galapagos*, 85.7% of the months showed any degree of positive precipitation anomaly (SPDI >1.0), and 100% of the months showed an extremely wet condition (SPDI ≥2.0), were associated to Year 2 of an extreme El Niño event (**Fig. 2**). Such extreme rainfall conditions were concomitant, with a lag of zero months, to the presence of warm SST temperatures in the easterly most Niño region (ie. Niño

1+2), and for both types, the EP El Niños (**Thielen et al., 2015; Bravo de Guenni et al., 2016; Morán-Tejeda et al., 2016; Quishpe-Vásquez et al., 2019**), and the coastal El Niño (**Thielen et al., 2021a; Rollenbeck et al., 2022**). Through transferring heat from the ocean to the atmosphere, this anomalous warming elevates air temperatures in the coastal region, triggering localized atmospheric convection and heavy rainfall (**Cai et al., 2020**). Based on SST at Niño 1+2 dynamics, **Thielen et al. (2016)** predicted that precipitation anomalies in the Ecuadorian coast generated by the mixed type El Niño 2015/16 would not

be as significant as those from the El Niño 82/83 and 97/98, forecast that was fully corroborated in the present research. This lack of response in coastal and insular precipitation is most certainly true for CP El Niño events, as well as the mixed (EP-CP) El Niño type, such as **MIX-EN 15/16**. As for the *Amazon slope*, even though the number of months showing any degree of positive precipitation anomaly (SPDI >1.0) doubled that of the coastal and insular zones, less than 4% of the months occurred during an extreme El Niño event, from which none reached the extremely wet condition (SPDI ≥2.0).

At the *Pacific slope*, there are no significant differences (*P*>0.05) on SPDI values resulting from **EP-EN 82/83** and **EP-EN 97/98**, when considered on annual bases. For both of these Eastern Pacific mega-El Niño events, precipitation anomalies lasted 10 months, reaching mean SPDI values of 2.09 for 1983, and 2.39 for 1998 (**Tables 2 and 3**). Differences between these two events become extremely significant when comparing the first 6 months - Precipitation anomalies during **EP-EN 97/98** occurred sooner and reached faster maximum SPDI values during the first 6 months of 1998 than during 1983, or any other

extreme El Niño event. SPDI dynamics for the first six months of Year 2 between **EP-EN 82/83** and **COA-EN 17** tended to be similar (P>0.01) (**Tables 2 and 3**).

Regarding how far the extreme El Niño events influence extends in the *Pacific slope*, the present study identified three most relevant facts:

1.  For any extreme El Niño event, over 50% of all extreme anomalies occurred at elevations under 150m (**Fig. 4**). This
represents over 40% of Ecuadorian coastal surface, and involves a most strategic zone for Ecuador since it comprises almost all lowland agriculture and aquaculture, which after petroleum oil production, represent the main activities generating export products. This zone also holds, besides a high density of rural population as well as numerous small to medium size towns, the largest city in Ecuador: Guayaquil, with a little more than 2.5 million inhabitants.

2.  The difference between extreme El Niño events was more significant (P<0.05) when considering how far into the
Andes the precipitation anomalies are perceived. For instance, during the long-lasting **EP-EN 97/98**, 80% of all extreme anomalies (SPDI ≥2.0) occurred at elevations up to 500m, while for the relatively less lasting extreme events, such as **COA-EN 17** and **EN 82/83**, this value was reached at altitudes much higher: at 800m and 1000m, respectively (**Fig. 4**).



3. The difference between the three extreme El Niño events disappears at around 3000m asl when reaching the accumulation of 97% of all extreme anomalies (SPDI ≥2.0). At an altitude of 4000m, all extreme El Niño events reach the mean (series 1981-2010), which is a little before all reach 100% at the maximum height of 4300masl (**Fig. 4**).

Several authors have also investigated the ENSO influence extends inside continental Ecuador. **Bendix and Bendix (2006)**, for instance, showed that positive rainfall anomalies during ENSO mainly affect the coastal plain of Ecuador to the western slope of the Andes at altitudes <1800m; while **Pineda et al. (2013)**, observed ENSO signals at locations as high as 2700m. Regarding the presence of ENSO signal at high altitudes in the Pacific slope, relief plays a twofold role in the control of ocean–atmospheric forcing: It can modulate the atmospheric circulation, leading to a dissipation of the signal, or can favor

meteorological processes, leading to enhancement of orographic precipitation (**Pineda et al., 2013**). Now, there is no easy answer about the difference between the results from such studies regarding how far the extreme El Niño events influence extends on the *Pacific slope*.

Preexisting studies limit their analysis to specific areas of Ecuador or may confront severe data limitations due to discontinuities in space and in time. Such limitations were overcome in the present study. **Figure 5** is the spatial representation,

for the entire territory of Ecuador, of the mean annual (Year 2) precipitation anomalies (as SPDI) generated by the most important extreme El Niño events since 1981, that is the mega-El Niños **EP-EN 82/83** and **EP-EN 97/98**, and the Coastal El Niño **COA-EN 17**. From this figure, it is evident that the ENSO signal is variable, not only along the lowlands of the *Pacific slope* but also along the highlands and the dorsal of the Andes. From north to south, the first half of the 1,030 km of the dominant orographic barrier of the dorsal of the Andes, does not show any effect or signal from ENSO – That is, no

precipitation anomalies are generated by extreme El Niño at the highest sections of the hydrographic systems 02-MIRA, 06-ESMERALDAS and part of 13-GUAYAS. From this point on, and for 320 km along the dorsal of the Andes, the highest sections of systems 13-GUAYAS, 15-CANAR, 16-NARANJAL PAGUA, and 17-JUBONES, showed moderate to high precipitation anomalies (SPDI 1.0 - 1.5) during an extreme El Niño event. But then again, in the last 165 km of the dorsal of the Andes, which correspond to the highest sections of the southernmost hydrographic system, 22-CHIRA, any ENSO signal

disappeared. At this point, there is no clear pattern regarding extreme El Niño events and the generation precipitation anomalies in the highest sections of the Andean Cordillera. In any case, the ENSO signal was observed at mean altitudes ranging from 3200 to 3900m. Other physical determinants such as distance to coastline and steepness of the Cordillera may play an important role in determining the degree of ENSO signal. Coincidentally, the aforementioned 320 km transect that did reach precipitation anomalies was located at a distance from the seashore of 120 km or less, also showing a dominant steep relief (**Figs. 1 and 5**).

Now, from the results of the spatial-temporal analysis of precipitation dynamics, it is evident that the degree of seasonality also conditions the magnitude of the ENSO signal in entire continental Ecuador. **Figure 6**, for instance, shows that it is in the most extremely seasonal regions ($\overline{SI}$ 1.0 - 1.2) where precipitation anomalies are the strongest, while regions with low or no seasonality ($\overline{SI}$ 0.0 – 0.6) show no precipitation anomaly during the event of an extreme El Niño. According to **Carréric et al., (2019)**, the strong EP El Niño events peak in boreal winter is extended by two months, which results in significantly more





events peaking in February–March–April, the season when the climatological Inter-Tropical Convergence Zone is at its southernmost location. The *Pacific slope* shows strong seasonality, while the *Amazon slope* exhibits mild to no seasonality and the Sierra with a moderate seasonality (**Tobar and Wyseure, 2017**).

From **Figures 5 and 6**, a list of the most vulnerable hydrographic systems regarding affectation in the event of an extreme El

Niño becomes easily identifiable. Besides the insular system of 24-GALAPAGOS, the other 13 are continental systems from the *Pacific slope* that show the highest seasonality in their precipitation: 19-ARENILLAS (SPDI 2.47 and $\overline{SI}$ 1.18), 08-JAMA (2.31 and 1.19), 18-SANTA ROSA (2.31 and 1.11), 09-CHONE (2.25 and 1.23), 12-ZAPOTAL (2.23 and 1.33), 16-NARANJAL PAGUA (2.22 and 1.10), 23-ISLA PUNA (2.22 and 1.27), 11-JIPIJAPA (2.19 and 1.27), 20-ZARUMILLA (2.17 and 1.19), 21-PUYANGO (2.16 and 1.19), 10-PORTOVIEJO (2.11 and 1.29), 14-TAURA (2.09 and 1.29), and 13-GUAYAS

(1.87 and 1.15).

## 5 Conclusion

The present study generates reliable information about the most relevant aspects of spatial-temporal extreme precipitation dynamics resulting from the various types of extreme El Niño events, including the Coastal El Niño. Information that becomes most valuable and highly strategic considering that these extreme climatic events are expected to double their occurrence in

the foreseeable future.

- For both, the *Pacific slope* and *Galapagos*, it is during the first half of the second year of an extreme El Niño event, coincidentally encompassing the rainy season, when most precipitation extremes occur, and it is during this time when any difference between extreme El Niño events become more evident.

- There was no evidence of any significant precipitation anomaly generated on Ecuadorian territory by the occurrences

of the mixed (EP-CP) type of extreme El Niño. Likewise, there was no evidence of any significant precipitation anomaly generated on the *Amazon slope* by the occurrences of any type of extreme El Niño: eastern Pacific, Central Pacific, mixed or Coastal.

- For any extreme El Niño event, over 50% of all extreme anomalies (SPDI ≥2.0) occurred at elevations under 150m. But, differences between events become significant when considering how far into the Andes the precipitation

anomalies are perceived. For instance, during **EP-EN 97/98**, 80% of all extreme anomalies occurred at elevations up to 500m, while for **COA-EN 17** and **EN 82/83**, this was 800m and 1000m, respectively. Any difference between extreme El Niño events disappears again around 3000 m asl, when accumulative extreme anomalies reach 97%. Finally, at an altitude of 4000m, all extreme El Niño events reach the historical mean (series 1981-2010).

- Nevertheless, the ENSO signal is variable, not only along the lowlands of the *Pacific slope* but also in the highlands

and along the dorsal of the Andes. Here, the ENSO signal can be observed, in continuous sections of several hundred kilometers, and at mean altitudes ranging from 3200 to 3900m. Other physical determinants such as distance to



coastline and steepness of the Cordillera may play an important role in determining the degree of ENSO signal on the Andean Cordillera.

- Finally, the degree of seasonality also conditions the magnitude of the ENSO signal in entire continental Ecuador: It is in the regions showing the highest seasonality index where the most severe precipitation anomalies from extreme El Niño events occur. In these terms, 13 hydrographic systems from the *Pacific slope* showing strong seasonality resulted to be the most vulnerable to extreme precipitations generated by extreme El Niño events. Both the north of Ecuador and the *Amazon slope* exhibits mild to no seasonality. Concomitantly, hydrographic systems from these regions show no significant precipitation anomalies regardless of the type or strength of El Niño event.

Results from present research allowed us to generate most valuable information regarding similarities and differences between the effects on precipitation from types of extreme El Niño events, as well as highlights spatially and quantitatively, those regions or hydrographic systems where most extreme precipitations anomalies are most likely to occur in the event of an extreme El Niño, either eastern Pacific or coastal El Niño. Access to information like this is most strategic when designing and incorporating disaster-risk analyses and policies (**Ward et al., 2014**). For instance, the **Figure 5** shows where the most negative direct effects from such anomalies are expected, as well as where such extreme events may exert strong and widespread influences on both flood hazard and risk. Because extreme El Niño events have some predictive capacity, mainly the eastern Pacific type, these specific results represent a solid contribution toward developing a risk-predictive model with applications for improved disaster planning (**Ward et al., 2014**). The results also provide solid and opportune evidence for identifying, in the context of global climate change scenarios, an increase in the frequency and intensity of extreme climatic events, the most appropriate management practices aimed to achieve sustainability of ongoing anthropogenic activities in one of the most climatic vulnerable regions from the Pacific coast of South America, either by adapting or mitigating the direct effects such as flooding and mudslides, as well as by reducing the risk of indirect effects such as the case of the emergence of important infectious diseases in a region that, historically and linked to the occurrence of extreme climatic events, has shown to be most vulnerable to significant epidemics of cholera, leptospirosis, dengue, chikungunya, zika, malaria, etc. (**OPS-OMS, 1999; The World Bank Group, 2011**), and more recently to COVID-19, one of the worst pandemics known by humankind in recent history, about which there is currently no clue about what to expect and to control the spread of such disease in the event of an extreme El Niño event, in the Ecuador or elsewhere.

## Author contributions

Conceptualization: DT, PRP, MLP. Data curation: DT, MM, WR, JIQ, IASW. Formal analysis: DT, PRP, MM, WR, JIQ. Funding acquisition: DT, MAAA, AQ. Investigation: DT, PRP, MLP. Methodology: DT, PRP, MLP. Project administration:



DT, AQ, MAAA. Resources: DT, AQ, MAAA. Software: DT, MM, WR, JIQ, GB, IASW. Supervision: PRP, GB. Validation: DT, PRP, MLP. Visualization: DT, PRP, MM, WR, JIQ, MLP, IASW. Writing – original draft: DT, PRP, MLP.

**Acknowledgments**

Grant from FONACIT from the Ministry of Science and Technology of Venezuela allowed the acquisition of specialized
hardware and fulfill other requirements for the present research. Grants from the Organization of American States by the Programa de Alianças para a Educação e a Capacitação (Bolsas Brasil - PAEC OEA-GCUB) were assigned to PRP and IASW. The access to GIS software was provided by Service Unit of Geographical System Information (UniSIG), Ecology Center of the Venezuelan Institute for Scientific Research (IVIC), Caracas, Venezuela. This work was also supported by the mining canon (002-2019-P.CO-UNAH/FOCAM).

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



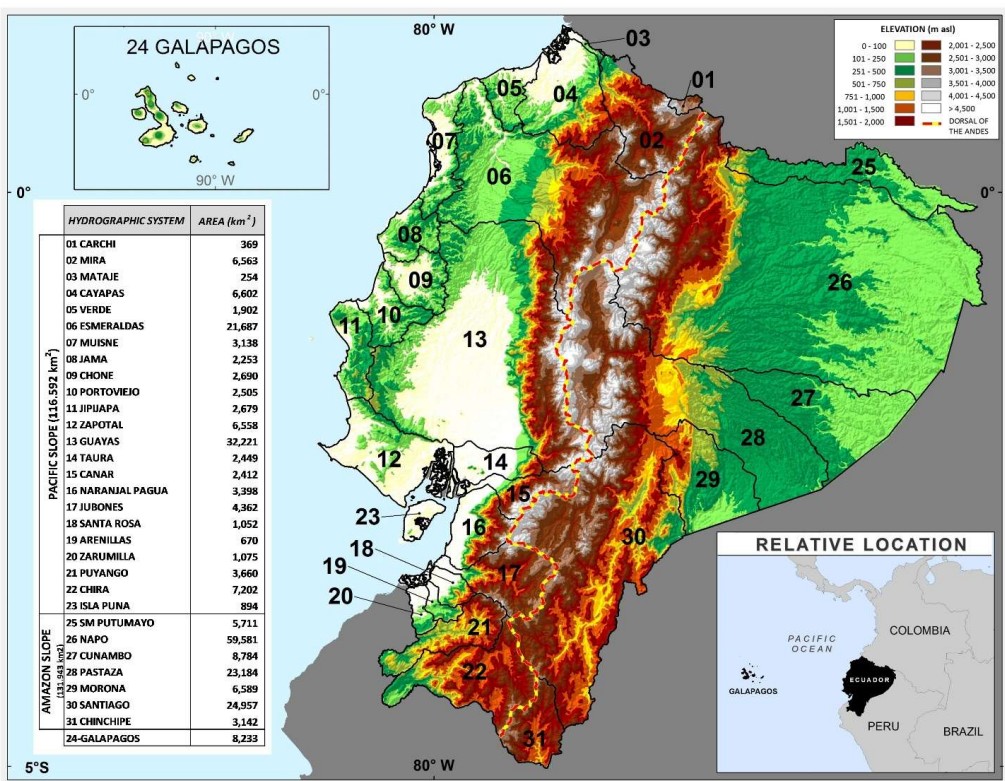

**Figure 1: Map of study area, defined as the totality of the territory of Ecuador. The continental territory is divided by the dorsal of the Andes into two main and distinctive zones: The *Pacific slope* (116,592 km2) and the *Amazon slope* (131,948 km2). Following CNRH (2002) classification system, each of these two continental zones was further divided into 30 hydrographic systems: 23 for the *Pacific slope*, and seven for the *Amazon slope*. Regarding the insular territory, a unique hydrographic system was established encompassing all offshore islands, specifically the *Galapagos* Islands (8,233 km2).**



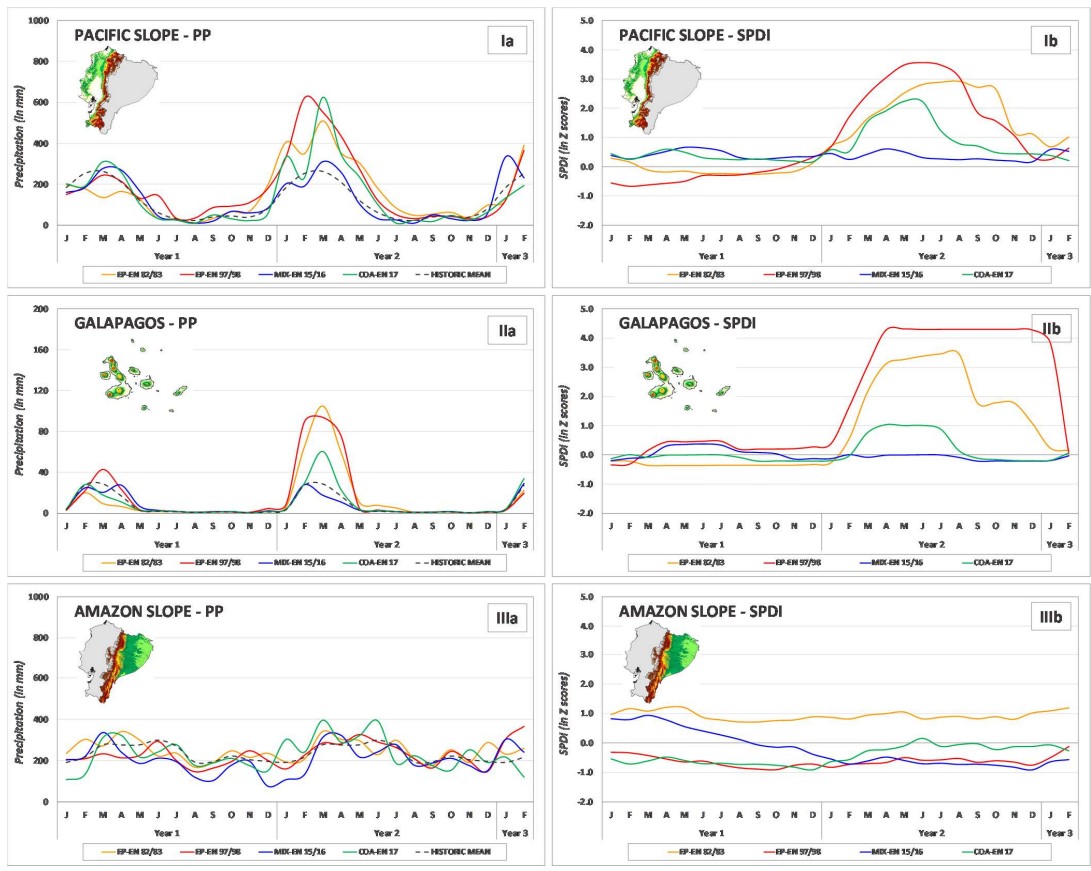

**Figure 2. (Ia, IIa, and IIIa): the mean monthly precipitation (in mm) for the main study zones (*Pacific slope*, *Galapagos*, and *Amazon slope*, correspondingly), for each extreme El Niño event (EP-EN 82/83, EP-EN 97/98, MIX-EN 15/16 and COA-EN 17) and Year 1 and Year 2. (Ib, IIb, and IIIb): the resulting mean Standardized Pluviometric Drought Index - SPDI (in Z scores).**



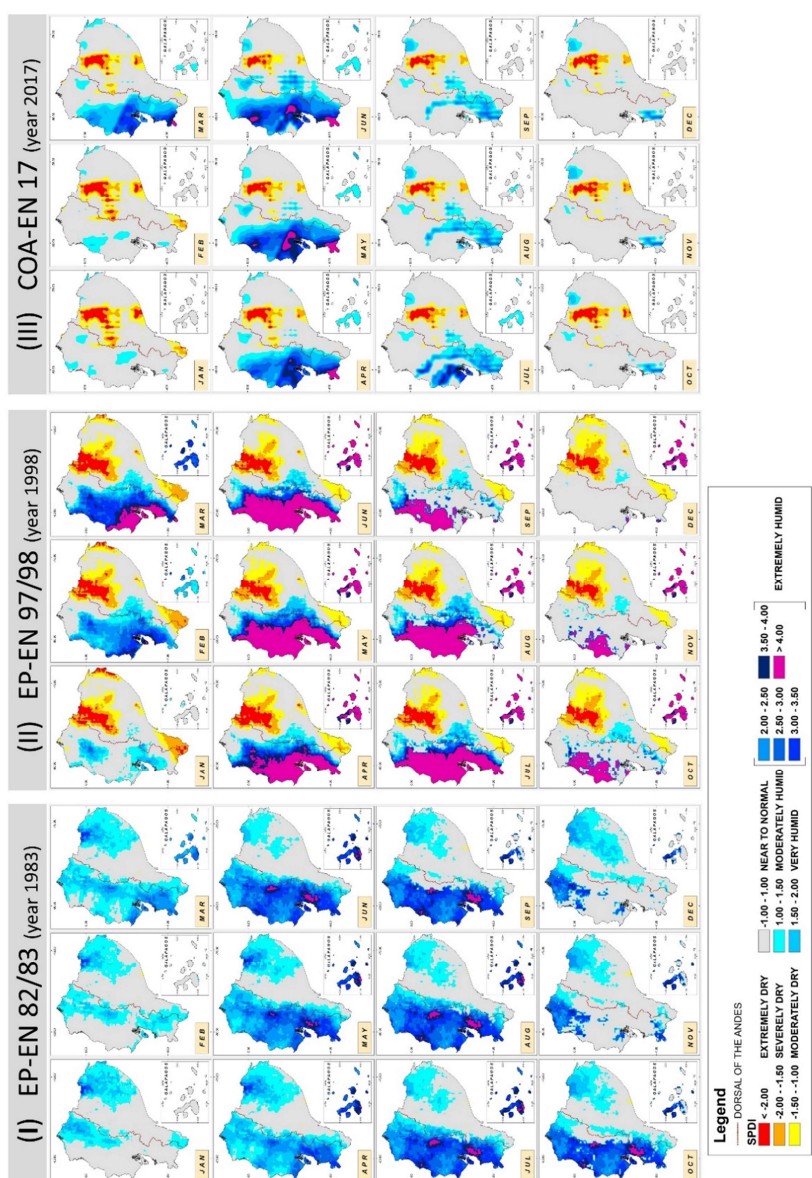

**Figure 3: Spatial dynamics of Standardized Pluviometric Drought Index (SPDI) of Year 2 for the mega-El Niño events EP-EN 82/83 (I) and EP-EN 97/98 (II), and the Coastal El Niño COA-EN 17 (III). SPDI categories are adapted from McKee et al. (1993).**





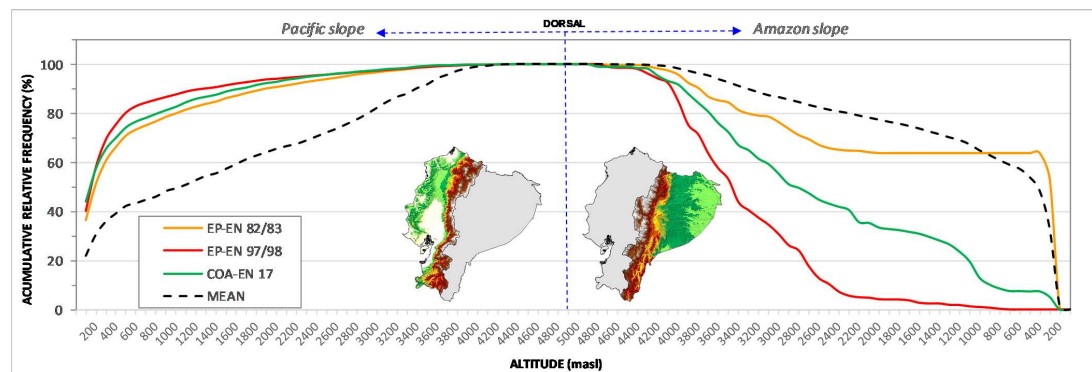

**Figure 4: Accumulative relative frequency (%) of extremely humid condition (SPDI ≥2.0) for the Year 2 of mega-El Niño events EP-EN 82/83 and EP-EN 97/98, and the Coastal El Niño COA-EN 17, regarding altitude (m asl) of continental Ecuador, as well as to the historic mean frequency (series 1981-2010).**



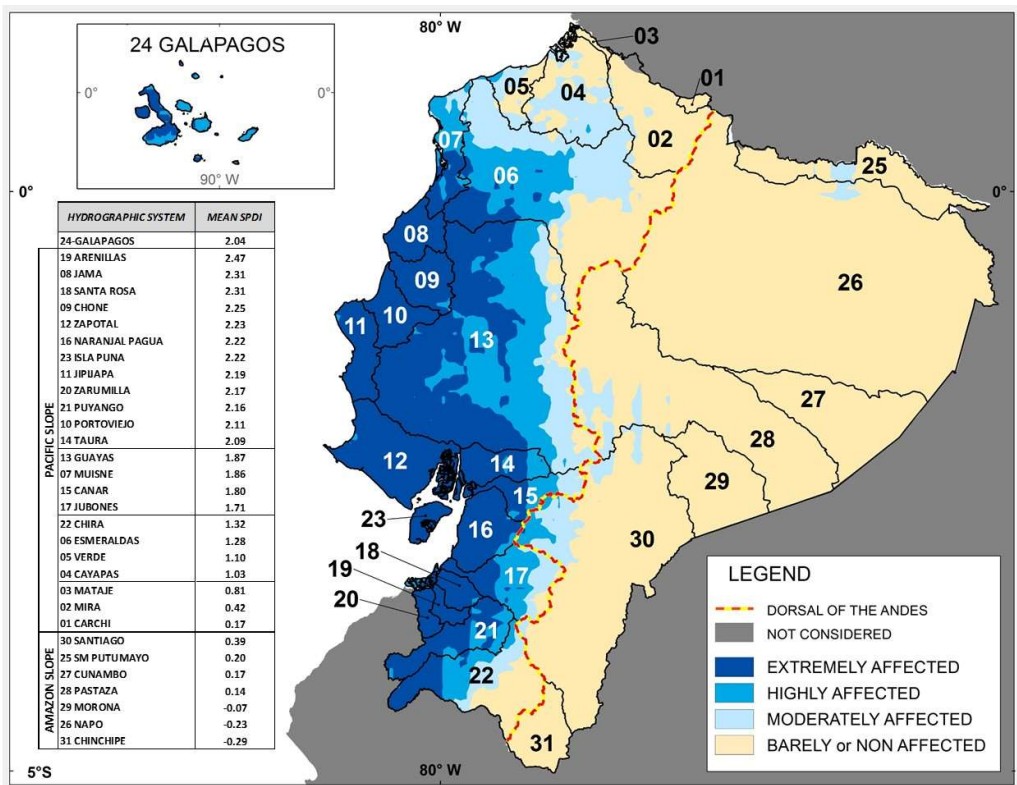

**Figure 5. Potential affectation from precipitation anomalies generated by extreme El Niño events, as determined from mean annual (Year 2) SPDI resulting from the mega-El Niño events EP-EN 82/83 and EP-EN 97/98, and the Coastal El Niño COA-EN 17.**





**Figure 6. Relationship at basin level between the Seasonality Index ($\overline{SI}$, Walsh and Lawler, 1981) and the mean annual (Year 2) SPDI resulting from the mega-El Niño events EP-EN 82/83 and EP-EN 97/98, and the Coastal El Niño COA-EN 17. Extremely humid condition (SPDI ≥2.0) is reached in basins with the highest $\overline{SI}$ values (ie. with rainy season concentrated in three or fewer months, $\overline{SI}$ ≥1.00).**



**Table 1. Categories resulting from SPDI estimation, adapted from McKee et al. (1993).**

| RANGE | CATEGORY |
|---|---|
| ≤ -2.00 | *Extremely dry* |
| -1.99 - -1.50 | *Very dry* |
| -1.49 - -1.00 | *Moderately dry* |
| -0.99 - 0.99 | *Near normal* |
| 1.00 - 1.49 | *Moderately humid* |
| 1.50 - 1.99 | *Very humid* |
| ≥ 2.00 | *Extremely humid* |



**Table 2.** Standardized Precipitation Drought Index (SPDI) temporal and spatial dynamics at the different hydrographic systems for 1983 (Year 2) of mega-El Niño event EP-EN 82/83. Cluster analysis (K-means clustering using Euclidean distance) was performed to both rows and columns, and with the statistical tool ClustVis (Metsalu and Vilo, 2015).

**(a) Monthly mean SPDI (in z score)**

| Hydrographic System | Row Cluster | JAN | FEB | MAR | APR | MAY | JUN | JUL | AUG | SEP | OCT | NOV | DEC |
|---|---|---|---|---|---|---|---|---|---|---|---|---|---|
| 16 NARANJAL PAGUA | 1 | 0,60 | 1,29 | 2,30 | 3,02 | 3,65 | 3,87 | 3,93 | 3,95 | 3,97 | 3,98 | 0,16 | 0,22 |
| 23 ISLA PUNA | | 0,11 | 1,50 | 2,40 | 2,85 | 3,16 | 3,31 | 3,35 | 3,36 | 3,37 | 3,38 | -0,11 | -0,09 |
| 17 JUBONES | | 0,51 | 1,31 | 1,97 | 2,35 | 2,81 | 2,91 | 2,96 | 2,96 | 3,02 | 3,06 | 0,60 | 0,69 |
| 14 TAURA | | 0,49 | 0,85 | 1,63 | 2,21 | 2,81 | 3,02 | 2,99 | 3,00 | 2,92 | 2,92 | 0,32 | 0,39 |
| 13 GUAYAS | | 0,62 | 0,82 | 1,39 | 1,85 | 2,49 | 2,81 | 2,86 | 2,89 | 2,48 | 2,46 | 0,71 | 0,79 |
| 15 CANAR | | 0,53 | 0,91 | 1,51 | 2,12 | 2,75 | 2,99 | 2,74 | 2,78 | 2,40 | 2,47 | 0,49 | 0,69 |
| 12 ZAPOTAL | | -0,13 | 0,80 | 2,04 | 2,37 | 2,67 | 2,95 | 3,00 | 3,01 | 3,03 | 2,95 | 0,73 | 0,74 |
| 21 PUYANGO | | 0,27 | 1,11 | 2,37 | 2,82 | 3,15 | 3,21 | 3,22 | 3,16 | 3,17 | 3,09 | 1,27 | 0,62 |
| 18 SANTA ROSA | | 0,22 | 1,17 | 2,27 | 2,75 | 3,19 | 3,33 | 3,39 | 3,39 | 3,41 | 2,65 | 0,25 | 0,27 |
| 19 ARENILLAS | | 0,23 | 1,09 | 2,33 | 2,76 | 3,14 | 3,26 | 3,29 | 3,30 | 3,31 | 2,38 | 0,47 | 0,49 |
| 20 ZARUMILLA | | -0,01 | 1,04 | 2,30 | 2,75 | 3,09 | 3,20 | 3,24 | 3,24 | 3,26 | 2,59 | 0,48 | 0,33 |
| 11 JIPIJAPA | 2 | -0,26 | 0,24 | 1,03 | 1,27 | 1,70 | 2,30 | 2,53 | 2,58 | 2,60 | 2,61 | 0,06 | 0,08 |
| 09 CHONE | | 0,02 | 0,33 | 1,34 | 2,02 | 2,55 | 3,10 | 3,32 | 3,34 | 3,37 | 2,79 | 0,37 | 0,40 |
| 10 PORTOVIEJO | | -0,13 | 0,24 | 1,22 | 1,73 | 2,20 | 2,73 | 2,93 | 2,98 | 2,99 | 2,73 | 0,15 | 0,18 |
| 07 MUISNE | | -0,09 | 0,13 | 0,69 | 1,14 | 1,73 | 2,34 | 2,75 | 2,88 | 2,96 | 3,08 | 1,53 | 0,95 |
| 08 JAMA | | -0,07 | 0,21 | 1,19 | 1,85 | 2,54 | 3,24 | 3,54 | 3,59 | 3,62 | 3,59 | 1,31 | 1,33 |
| 29 MORONA | 3 | 0,10 | -0,07 | 0,09 | 0,08 | 0,28 | 0,19 | 0,24 | 0,37 | 0,25 | 0,29 | 0,19 | 0,82 |
| 28 PASTAZA | | 0,63 | 0,57 | 0,66 | 0,68 | 0,86 | 0,81 | 0,96 | 1,06 | 0,61 | 0,73 | 0,63 | 0,99 |
| 30 SANTIAGO | | 0,28 | 0,40 | 0,58 | 0,68 | 0,97 | 0,85 | 0,77 | 0,76 | 0,76 | 0,85 | 0,56 | 0,78 |
| 31 CHINCHIPE | 4 | 1,26 | 1,22 | 1,34 | 1,09 | 1,06 | 1,09 | 0,94 | 0,88 | 0,88 | 0,98 | 0,92 | 0,86 |
| 26 NAPO | | 1,14 | 1,06 | 1,18 | 1,26 | 1,18 | 0,85 | 0,94 | 0,95 | 0,95 | 0,99 | 0,97 | 1,08 |
| 27 CUNAMBO | | 1,19 | 0,90 | 1,12 | 1,13 | 1,14 | 0,82 | 0,81 | 0,81 | 0,73 | 0,87 | 0,75 | 1,06 |
| 25 SM PUTUMAYO | | 1,62 | 1,55 | 1,65 | 1,76 | 1,68 | 1,20 | 1,25 | 1,33 | 1,37 | 1,46 | 1,56 | 1,64 |
| 01 CARCHI | | 1,51 | 1,46 | 1,56 | 1,64 | 1,73 | 1,77 | 1,64 | 1,64 | 1,50 | 1,46 | 1,27 | 1,39 |
| 02 MIRA | | 1,54 | 1,16 | 1,32 | 1,54 | 1,66 | 1,67 | 1,49 | 1,54 | 1,35 | 1,36 | 1,31 | 1,42 |
| 05 VERDE | | 0,80 | 0,93 | 1,33 | 1,57 | 1,77 | 2,08 | 2,28 | 2,40 | 2,52 | 2,66 | 2,66 | 2,18 |
| 03 MATAJE | | 1,38 | 1,46 | 1,60 | 1,73 | 1,78 | 1,88 | 1,97 | 2,14 | 2,26 | 2,40 | 2,46 | 2,49 |
| 04 CAYAPAS | | 1,28 | 1,28 | 1,53 | 1,72 | 1,92 | 2,09 | 2,22 | 2,34 | 2,63 | 2,68 | 2,71 | |
| 06 ESMERALDAS | | 1,06 | 0,94 | 1,30 | 1,48 | 1,89 | 2,21 | 2,37 | 2,44 | 2,07 | 1,94 | 1,62 | 1,60 |
| 22 CHIRA | | 0,56 | 1,07 | 1,88 | 2,16 | 2,32 | 2,35 | 2,31 | 2,11 | 2,12 | 2,12 | 1,60 | 0,72 |

**(b) Relative area of basin with SPDI >2.0 (in %)**

| Hydrographic System | JAN | FEB | MAR | APR | MAY | JUN | JUL | AUG | SEP | OCT | NOV | DEC |
|---|---|---|---|---|---|---|---|---|---|---|---|---|
| 16 NARANJAL PAGUA | 0,0 | 0,0 | 84,7 | 100 | 100 | 100 | 100 | 100 | 100 | 100 | 0,0 | 0,0 |
| 23 ISLA PUNA | 0,0 | 0,0 | 100 | 100 | 100 | 100 | 100 | 100 | 100 | 100 | 0,0 | 0,0 |
| 17 JUBONES | 0,0 | 0,0 | 45,0 | 67,9 | 88,6 | 89,3 | 91,4 | 90,7 | 92,9 | 94,3 | 0,0 | 0,0 |
| 14 TAURA | 0,0 | 0,0 | 13,6 | 72,8 | 100 | 100 | 97,5 | 93,8 | 93,8 | | 0,0 | 0,0 |
| 13 GUAYAS | 0,1 | 1,4 | 9,8 | 36,0 | 85,0 | 97,5 | 94,8 | 95,0 | 79,3 | 77,8 | 12,1 | 12,1 |
| 15 CANAR | 0,0 | 0,0 | 19,0 | 63,3 | 82,3 | 86,1 | 73,4 | 74,7 | 58,2 | 58,2 | 0,0 | 0,0 |
| 12 ZAPOTAL | 0,0 | 0,0 | 52,4 | 84,3 | 96,7 | 100 | 100 | 100 | 100 | 97,6 | 0,0 | 0,0 |
| 21 PUYANGO | 0,0 | 0,0 | 96,7 | 100 | 100 | 100 | 100 | 97,5 | 97,5 | 94,2 | 30,0 | 6,7 |
| 18 SANTA ROSA | 0,0 | 0,0 | 82,9 | 100 | 100 | 100 | 100 | 100 | 100 | 77,1 | 0,0 | 0,0 |
| 19 ARENILLAS | 0,0 | 0,0 | 100 | 100 | 100 | 100 | 100 | 100 | 100 | 68,2 | 4,5 | 4,5 |
| 20 ZARUMILLA | 0,0 | 0,0 | 100 | 100 | 100 | 100 | 100 | 100 | 100 | 80,0 | 11,4 | 5,7 |
| 11 JIPIJAPA | 0,0 | 0,0 | 0,0 | 13,6 | 75,0 | 84,1 | 86,4 | 88,6 | 88,6 | | | |
| 09 CHONE | 0,0 | 0,0 | 52,4 | 100 | 100 | 100 | 100 | 100 | 82,1 | | 2,4 | 2,4 |
| 10 PORTOVIEJO | 0,0 | 0,0 | 10,6 | 89,4 | 100 | 100 | 100 | 100 | 91,8 | | 0,0 | 0,0 |
| 07 MUISNE | 0,0 | 0,0 | 21,6 | 69,1 | 97,9 | 100 | 100 | 100 | | | 47,4 | 21,6 |
| 08 JAMA | 4,3 | 37,1 | 94,3 | 100 | 100 | 100 | 100 | 98,6 | | | 32,9 | 32,9 |
| 29 MORONA | 0,0 | 0,0 | 0,0 | 0,0 | 0,0 | 0,0 | 0,0 | 0,0 | 0,0 | 0,0 | 0,0 | 0,0 |
| 28 PASTAZA | 0,0 | 0,0 | 0,1 | 3,9 | 5,6 | 10,4 | 12,0 | 0,0 | 0,0 | | | |
| 30 SANTIAGO | 0,0 | 0,0 | 0,5 | 3,3 | 11,6 | 12,4 | 10,1 | 8,0 | 8,3 | 8,3 | | |
| 31 CHINCHIPE | 0,0 | 0,0 | 1,9 | 0,0 | 0,0 | 0,0 | | | | | | |
| 26 NAPO | 5,2 | 3,8 | 7,2 | 9,9 | 9,4 | 0,0 | 0,3 | 0,5 | 0,7 | 2,0 | 3,2 | 4,0 |
| 27 CUNAMBO | 0,7 | 0,0 | | | | | | | | | | |
| 25 SM PUTUMAYO | 25,6 | 20,5 | 24,4 | 25,0 | 25,6 | 0,0 | 4,5 | 6,3 | 8,0 | 24,4 | 36,4 | 34,7 |
| 01 CARCHI | 0,0 | 0,0 | 0,0 | 9,1 | 0,0 | 0,0 | | | | | | |
| 02 MIRA | 17,2 | 1,0 | 2,9 | 6,9 | 13,2 | 22,5 | 15,7 | 19,1 | 6,9 | 10,8 | 16,2 | 20,1 |
| 05 VERDE | 0,0 | 0,0 | 0,0 | 13,6 | 20,3 | 49,2 | 71,2 | 88,1 | 96,6 | 100 | 98,3 | 79,7 |
| 03 MATAJE | 0,0 | 0,0 | 0,0 | 0,0 | 22,2 | 66,7 | 66,7 | 100 | 100 | 100 | 100 | |
| 04 CAYAPAS | 0,0 | 0,0 | 2,3 | 8,8 | 34,0 | 67,0 | 81,4 | 90,7 | 96,7 | 98,6 | 98,6 | 97,2 |
| 06 ESMERALDAS | 4,2 | 1,0 | 7,0 | 12,0 | 37,7 | 62,6 | 72,0 | 74,1 | 59,8 | 52,4 | 41,0 | 38,4 |
| 22 CHIRA | 0,0 | 0,9 | 45,1 | 54,0 | 57,5 | 58,0 | 58,0 | 47,8 | 47,8 | 48,2 | 33,2 | 1,3 |

**LEGEND (a)**

| | |
|---|---|
| ≤ -1.50 | SEVERELY DRY |
| -1.49 - -1.00 | MODERATELY DRY |
| -0.99 - 0.99 | NEAR NORMAL |
| 1.00 - 1.49 | MODERATELY HUMID |
| 1.50 - 1.99 | SEVERELY HUMID |
| ≥ 2.00 | EXTREMELY HUMID |

**LEGEND (b)**

| |
|---|
| 0.0% |
| 0.1 - 19.9% |
| 20.0 - 39.9% |
| 40.0 - 59.9% |
| 60.0 - 99.9% |
| 100% |





**Table 3.** Standardized Precipitation Drought Index (SPDI) temporal and spatial dynamics at the different hydrographic systems for 1998 (Year 2) of mega-El Niño event EP-EN 97/98. Cluster analysis (K-means clustering using Euclidean distance) was performed on both rows and columns and with the statistical tool ClustVis (Metsalu and Vilo, 2015).

| Hydrographic System | Row Cluster | (a) Monthly mean SPDI (in z score) | | | | | | | | | | | | (b) Relative area of basin with SPDI >2.0 (in %) | | | | | | | | | | | |
|---|---|---|---|---|---|---|---|---|---|---|---|---|---|---|---|---|---|---|---|---|---|---|---|---|---|
| | | JAN | FEB | MAR | APR | MAY | JUN | JUL | AUG | SEP | OCT | NOV | DEC | JAN | FEB | MAR | APR | MAY | JUN | JUL | AUG | SEP | OCT | NOV | DEC |
| 17 JUBONES | 1 | 0,44 | 1,66 | 2,25 | 2,81 | 3,24 | 3,05 | 3,07 | 1,70 | 0,95 | 1,01 | 0,59 | 0,59 | 0,0 | 40,0 | 51,4 | 64,3 | 78,6 | 70,0 | 70,0 | 35,0 | 12,1 | 12,9 | 0,0 | 0,0 |
| 06 ESMERALDAS | 1 | 0,85 | 1,13 | 1,46 | 1,70 | 2,07 | 2,23 | 2,00 | 2,04 | 1,78 | 1,36 | 0,84 | 0,50 | 8,7 | 22,5 | 37,0 | 40,3 | 45,9 | 50,1 | 42,3 | 41,9 | 35,0 | 19,7 | 6,0 | 0,0 |
| 21 PUYANGO | 1 | 0,92 | 2,29 | 3,26 | 4,09 | 4,36 | 4,37 | 4,37 | 1,23 | 1,20 | 1,22 | 0,82 | 0,85 | 0,0 | 65,0 | 97,5 | 100 | 100 | 100 | 100 | 24,2 | 23,3 | 23,3 | 10,0 | 10,0 |
| 08 JAMA | 1 | 0,75 | 1,78 | 3,33 | 4,21 | 4,74 | 5,15 | 5,20 | 4,01 | 3,09 | 1,67 | | 0,26 | 0,0 | 28,6 | 100 | 100 | 100 | 100 | 100 | 100 | 77,1 | 58,6 | 28,6 | 0,0 |
| 07 MUISNE | 1 | 1,02 | 1,86 | 2,77 | 3,57 | 4,18 | 4,70 | 4,85 | 4,70 | 4,79 | 2,65 | 0,60 | 0,28 | 3,1 | 43,3 | 87,6 | 97,9 | 99,0 | 100 | 100 | 96,9 | 96,9 | 47,4 | 5,2 | 0,0 |
| 13 GUAYAS | 1 | 0,91 | 2,10 | 2,93 | 3,54 | 4,13 | 4,15 | 3,96 | 3,95 | 2,19 | 1,94 | 1,25 | 0,33 | 2,6 | 62,0 | 83,2 | 86,1 | 97,4 | 93,2 | 86,6 | 85,9 | 38,5 | 33,3 | 18,1 | 0,0 |
| 16 NARANJAL PAGUA | 1 | 1,67 | 3,04 | 3,79 | 4,68 | 5,08 | 5,09 | 5,11 | 4,36 | 0,32 | 0,34 | 0,17 | 0,16 | 17,1 | 100 | 100 | 100 | 100 | 100 | 100 | 87,4 | 3,6 | 3,6 | 0,0 | 0,0 |
| 14 TAURA | 1 | 1,30 | 2,72 | 3,53 | 4,46 | 4,95 | 4,98 | 4,99 | 4,84 | 0,32 | 0,32 | 0,31 | 0,29 | 0,0 | 100 | 100 | 100 | 100 | 100 | 100 | 96,3 | 0,0 | 0,0 | 0,0 | 0,0 |
| 15 CANAR | 1 | 1,24 | 2,34 | 2,85 | 3,46 | 4,10 | 3,79 | 3,82 | 3,77 | 0,59 | 0,62 | 0,49 | 0,43 | 0,0 | 77,2 | 88,6 | 97,5 | 100 | 87,3 | 87,3 | 84,8 | 2,5 | 2,5 | 0,0 | 0,0 |
| 12 ZAPOTAL | 1 | 1,00 | 3,28 | 4,73 | 5,35 | 5,59 | 5,62 | 5,63 | 5,63 | 1,53 | 1,28 | 0,91 | 0,40 | 0,0 | 100 | 100 | 100 | 100 | 100 | 100 | 100 | 29,0 | 24,8 | 17,6 | 8,1 |
| 19 ARENILLAS | 1 | 1,35 | 3,13 | 4,56 | 5,35 | 5,62 | 5,63 | 5,63 | 0,55 | 0,33 | 0,35 | 0,33 | 0,35 | 0,0 | 100 | 100 | 100 | 100 | 100 | 100 | 4,5 | 0,0 | 0,0 | 0,0 | 0,0 |
| 18 SANTA ROSA | 1 | 1,63 | 3,30 | 4,71 | 5,48 | 5,76 | 5,77 | 5,78 | 1,03 | 0,26 | 0,27 | 0,25 | 0,27 | 2,9 | 100 | 100 | 100 | 100 | 100 | 100 | 14,3 | 0,0 | 0,0 | 0,0 | 0,0 |
| 20 ZARUMILLA | 1 | 1,18 | 3,24 | 4,75 | 5,65 | 5,94 | 5,96 | 5,96 | 0,16 | 0,16 | 0,16 | 0,15 | 0,17 | 0,0 | 100 | 100 | 100 | 100 | 100 | 100 | 0,0 | 0,0 | 0,0 | 0,0 | 0,0 |
| 23 ISLA PUNA | 1 | 1,46 | 3,76 | 4,97 | 5,83 | 6,06 | 6,08 | 6,08 | 2,29 | 0,40 | 0,40 | 0,05 | 0,05 | 0,0 | 100 | 100 | 100 | 100 | 100 | 100 | 40,6 | 9,4 | 9,4 | 3,1 | 3,1 |
| 10 PORTOVIEJO | 2 | 0,34 | 1,91 | 3,19 | 4,27 | 4,67 | 4,86 | 4,88 | 4,89 | 4,90 | 4,84 | 4,85 | 0,14 | 0,0 | 40,0 | 100 | 100 | 100 | 100 | 100 | 100 | 100 | 98,8 | 98,8 | 0,0 |
| 09 CHONE | 2 | 0,58 | 1,59 | 3,07 | 4,13 | 4,58 | 4,89 | 4,91 | 4,92 | 4,52 | 4,10 | 3,89 | 0,25 | 0,0 | 9,5 | 100 | 100 | 100 | 100 | 100 | 100 | 91,7 | 83,3 | 78,6 | 0,0 |
| 11 JIPIJAPA | 2 | 0,87 | 2,92 | 4,20 | 4,70 | 5,04 | 5,18 | 5,20 | 5,21 | 5,22 | 5,23 | 5,24 | 0,06 | 0,0 | 88,6 | 100 | 100 | 100 | 100 | 100 | 100 | 100 | 100 | 100 | 0,0 |
| 03 MATAJE | 3 | -0,90 | -0,74 | -0,56 | -0,38 | -0,35 | -0,18 | 0,05 | 0,10 | 0,17 | 0,08 | 0,07 | -0,01 | 0,0 | 0,0 | 0,0 | 0,0 | 0,0 | 0,0 | 0,0 | 0,0 | 0,0 | 0,0 | 0,0 | 0,0 |
| 02 MIRA | 3 | -0,56 | -0,56 | -0,60 | -0,58 | -0,34 | -0,29 | -0,29 | -0,18 | -0,20 | -0,22 | -0,18 | -0,28 | 0,0 | 0,0 | 0,0 | 0,0 | 0,5 | 0,5 | 0,5 | 1,0 | 1,0 | 0,0 | 0,0 | 0,0 |
| 29 MORONA | 3 | -0,27 | -0,14 | -0,20 | -0,29 | 0,05 | 0,05 | 0,22 | 0,42 | 0,33 | 0,37 | 0,26 | 0,11 | 0,0 | 0,0 | 0,0 | 0,0 | 0,0 | 0,0 | 0,0 | 1,9 | 0,0 | 0,0 | 0,0 | 0,0 |
| 28 PASTAZA | 3 | -0,29 | -0,11 | -0,09 | -0,08 | 0,25 | 0,12 | 0,15 | 0,25 | 0,07 | 0,01 | 0,00 | -0,19 | 0,0 | 0,7 | 0,4 | 0,9 | 11,3 | 7,3 | 7,7 | 8,4 | 2,7 | 2,7 | 0,0 | 0,0 |
| 27 CUNAMBO | 3 | -0,33 | -0,37 | -0,34 | -0,33 | -0,26 | -0,32 | -0,40 | -0,42 | -0,45 | -0,42 | -0,48 | -0,61 | 0,0 | 0,0 | 0,0 | 0,0 | 0,0 | 0,0 | 0,0 | 0,0 | 0,0 | 0,0 | 0,0 | 0,0 |
| 30 SANTIAGO | 3 | -0,10 | 0,30 | 0,33 | 0,43 | 0,85 | 0,52 | 0,61 | 0,71 | 0,37 | 0,42 | 0,26 | 0,18 | 1,2 | 6,9 | 8,8 | 10,0 | 20,2 | 10,3 | 14,4 | 17,5 | 4,0 | 3,0 | 0,0 | 0,0 |
| 04 CAYAPAS | 3 | -0,31 | -0,14 | 0,09 | 0,29 | 0,44 | 0,65 | 0,82 | 0,88 | 0,94 | 0,47 | 0,44 | 0,34 | 0,0 | 0,0 | 0,0 | 0,0 | 0,9 | 1,9 | 1,9 | 2,3 | 2,8 | 0,0 | 0,0 | 0,0 |
| 05 VERDE | 3 | -0,55 | -0,20 | 0,24 | 0,63 | 0,88 | 1,22 | 1,44 | 1,47 | 1,57 | 0,58 | 0,57 | 0,50 | 0,0 | 0,0 | 0,0 | 6,8 | 11,9 | 20,3 | 23,7 | 23,7 | 25,4 | 0,0 | 0,0 | 0,0 |
| 22 CHIRA | 3 | -0,69 | 0,05 | 0,80 | 1,47 | 1,64 | 1,64 | 1,64 | 0,52 | 0,52 | 0,56 | 0,22 | 0,20 | 0,0 | 9,7 | 30,1 | 35,0 | 38,5 | 38,5 | 38,5 | 9,3 | 8,8 | 9,3 | 0,4 | 0,4 |
| 26 NAPO | 4 | -1,32 | -1,35 | -1,32 | -1,29 | -1,28 | -1,30 | -1,32 | -1,32 | -1,38 | -1,34 | -1,30 | -1,37 | 0,0 | 0,0 | 0,0 | 0,0 | 0,0 | 0,0 | 0,0 | 0,0 | 0,0 | 0,0 | 0,0 | 0,0 |
| 25 SM PUTUMAYO | 4 | -1,28 | -1,31 | -1,29 | -1,28 | -1,23 | -1,10 | -1,15 | -1,12 | -1,20 | -1,20 | -1,14 | -1,18 | 0,0 | 0,0 | 0,0 | 0,0 | 0,0 | 0,0 | 0,0 | 0,0 | 0,0 | 0,0 | 0,0 | 0,0 |
| 31 CHINCHIPE | 4 | -1,95 | -1,83 | -1,66 | -1,42 | -1,22 | -1,24 | -1,23 | -1,24 | -1,24 | -1,01 | -1,01 | -1,15 | 0,0 | 0,0 | 0,0 | 0,0 | 0,0 | 0,0 | 0,0 | 0,0 | 0,0 | 0,0 | 0,0 | 0,0 |
| 01 CARCHI | 4 | -1,16 | -1,17 | -1,26 | -1,35 | -1,07 | -1,08 | -1,07 | -0,90 | -0,92 | -0,80 | -0,72 | -0,82 | 0,0 | 0,0 | 0,0 | 0,0 | 0,0 | 0,0 | 0,0 | 0,0 | 0,0 | 0,0 | 0,0 | 0,0 |

**LEGEND (a)**

| | |
|---|---|
| ≤ -1.50 | SEVERELY DRY |
| -1.49 - -1.00 | MODERATELY DRY |
| -0.99 - 0.99 | NEAR NORMAL |
| 1.00 - 1.49 | MODERATELY HUMID |
| 1.50 - 1.99 | SEVERELY HUMID |
| ≥ 2.00 | EXTREMELY HUMID |

**LEGEND (b)**

| |
|---|
| 0.0% |
| 0.1 - 19.9% |
| 20.0 - 39.9% |
| 40.0 - 59.9% |
| 60.0 - 99.9% |
| 100% |





**Table 4.** Standardized Precipitation Drought Index (SPDI) temporal and spatial dynamics at the different hydrographic systems for year 2017 of Coastal El Niño (COA-EN 17). Cluster analysis (K-means clustering using Euclidean distance) was performed on both rows and columns and with the statistical tool ClustVis (Metsalu and Vilo, 2015).

### (a) Monthly mean SPDI (in z score)

| Hydrographic System | Row Cluster | JAN | FEB | MAR | APR | MAY | JUN | JUL | AUG | SEP | OCT | NOV | DEC |
|---|---|---|---|---|---|---|---|---|---|---|---|---|---|
| 20 ZARUMILLA | 1 | 0,59 | 0,79 | 2,85 | 3,37 | 3,57 | 3,58 | 0,70 | 0,70 | 0,71 | 0,71 | 0,71 | 0,73 |
| 14 TAURA | | 0,87 | 0,50 | 2,63 | 3,28 | 3,57 | 3,61 | 1,25 | 1,03 | 1,03 | 0,32 | 0,31 | 0,34 |
| 09 CHONE | | 0,73 | 1,10 | 1,98 | 2,61 | 3,13 | 3,24 | 1,75 | 0,58 | 0,58 | 0,29 | 0,29 | 0,31 |
| 08 JAMA | | 1,00 | 1,37 | 2,32 | 2,88 | 3,65 | 3,79 | 1,47 | 0,29 | 0,30 | 0,29 | 0,28 | 0,29 |
| 16 NARANJAL PAGUA | | 0,62 | 0,28 | 1,79 | 2,42 | 2,73 | 2,78 | 1,30 | 0,99 | 0,99 | 0,65 | 0,32 | 0,35 |
| 15 CANAR | | 0,52 | 0,11 | 1,62 | 2,04 | 2,39 | 2,56 | 1,39 | 1,36 | 1,31 | 0,63 | 0,45 | 0,49 |
| 13 GUAYAS | | 0,65 | 0,57 | 1,55 | 2,09 | 2,48 | 2,46 | 1,41 | 0,74 | 0,63 | 0,38 | 0,34 | 0,36 |
| 17 JUBONES | | 0,59 | 0,44 | 1,40 | 1,72 | 1,99 | 2,11 | 1,69 | 1,53 | 1,55 | 0,69 | 0,67 | 0,69 |
| 11 JIPIJAPA | | 0,32 | 0,66 | 2,25 | 2,40 | 2,66 | 2,19 | 2,11 | 0,08 | 0,05 | 0,06 | 0,05 | 0,05 |
| 10 PORTOVIEJO | | 0,52 | 0,90 | 1,80 | 2,13 | 2,48 | 2,55 | 1,24 | 0,15 | 0,15 | 0,15 | 0,14 | 0,15 |
| 12 ZAPOTAL | | 0,19 | 0,53 | 3,05 | 3,36 | 3,56 | 2,11 | 2,06 | 0,42 | 0,05 | 0,00 | -0,01 | 0,00 |
| 23 ISLA PUNA | | 0,21 | 0,43 | 2,22 | 2,81 | 2,98 | 2,99 | 2,99 | 0,94 | 0,53 | -0,01 | -0,11 | -0,10 |
| 22 CHIRA | | 0,60 | 0,64 | 2,18 | 2,50 | 2,60 | 2,51 | 1,35 | 1,36 | 1,23 | 0,94 | 0,93 | 0,91 |
| 21 PUYANGO | | 0,73 | 0,73 | 2,43 | 2,99 | 3,16 | 3,17 | 1,69 | 1,70 | 1,70 | 0,94 | 0,94 | 0,96 |
| 18 SANTA ROSA | | 0,55 | 0,58 | 2,25 | 2,69 | 2,89 | 2,92 | 2,16 | 2,11 | 2,12 | 1,35 | 1,36 | 1,38 |
| 05 VERDE | 2 | 0,75 | 0,48 | 0,52 | 0,65 | 0,91 | 1,16 | 0,55 | 0,59 | 0,58 | 0,67 | 0,62 | 0,59 |
| 04 CAYAPAS | | 0,60 | 0,43 | 0,49 | 0,58 | 0,80 | 0,97 | 0,51 | 0,55 | 0,51 | 0,59 | 0,56 | 0,50 |
| 03 MATAJE | | 0,62 | 0,50 | 0,56 | 0,64 | 0,85 | 0,95 | 0,68 | 0,72 | 0,68 | 0,71 | 0,69 | 0,63 |
| 06 ESMERALDAS | | 0,50 | 0,33 | 0,66 | 0,76 | 1,03 | 1,24 | 0,48 | 0,50 | 0,47 | 0,46 | 0,43 | 0,41 |
| 07 MUISNE | | 0,94 | 0,73 | 1,10 | 1,47 | 2,12 | 2,39 | 0,32 | 0,33 | 0,33 | 0,36 | 0,34 | 0,34 |
| 25 SM PUTUMAYO | | -0,02 | 0,06 | 0,20 | 0,20 | 0,27 | 0,41 | 0,28 | 0,37 | 0,46 | 0,35 | 0,49 | 0,47 |
| 02 MIRA | | 0,01 | -0,05 | 0,16 | 0,14 | 0,28 | 0,39 | 0,27 | 0,31 | 0,23 | 0,22 | 0,13 | 0,07 |
| 27 CUNAMBO | | -0,29 | -0,10 | 0,07 | -0,01 | 0,15 | -0,09 | | -0,07 | -0,02 | -0,18 | 0,05 | 0,15 |
| 01 CARCHI | | -0,28 | -0,29 | -0,01 | -0,10 | 0,10 | 0,16 | 0,13 | 0,14 | 0,09 | 0,00 | -0,08 | -0,09 |
| 30 SANTIAGO | | -0,32 | -0,33 | 0,20 | 0,39 | 0,66 | 1,07 | 0,69 | 0,70 | 0,72 | 0,46 | 0,44 | 0,46 |
| 29 MORONA | 3 | -1,09 | -1,07 | -0,83 | -0,59 | -0,41 | -0,07 | -0,40 | -0,30 | -0,26 | -0,46 | -0,40 | -0,32 |
| 28 PASTAZA | | -1,02 | -0,95 | -0,57 | -0,50 | -0,27 | 0,18 | -0,17 | -0,07 | -0,05 | -0,37 | -0,29 | -0,24 |
| 26 NAPO | | -0,65 | -0,56 | -0,37 | -0,42 | -0,36 | -0,21 | -0,43 | -0,36 | -0,33 | -0,49 | -0,34 | -0,37 |
| 31 CHINCHIPE | | -1,43 | -1,46 | -0,63 | -0,46 | -0,32 | -0,25 | -0,28 | -0,26 | -0,28 | -0,38 | -0,46 | -0,61 |
| 19 ARENILLAS | 4 | 0,88 | 0,97 | 2,75 | 3,27 | 3,49 | 3,51 | 2,50 | 2,51 | 2,51 | 2,39 | 2,40 | 2,42 |

### (b) Relative area of basin with SPDI >2.0 (in %)

| Hydrographic System | JAN | FEB | MAR | APR | MAY | JUN | JUL | AUG | SEP | OCT | NOV | DEC |
|---|---|---|---|---|---|---|---|---|---|---|---|---|
| 20 ZARUMILLA | 0,0 | 0,0 | 100 | 100 | 100 | 100 | 14,3 | 14,3 | 14,3 | 14,3 | 14,3 | 14,3 |
| 14 TAURA | 0,0 | 0,0 | 95,1 | 100 | 100 | 100 | 32,1 | 25,9 | 25,9 | 0,0 | 0,0 | 0,0 |
| 09 CHONE | 0,0 | 0,0 | 42,9 | 96,4 | 100 | 100 | 46,4 | 9,5 | 9,5 | 0,0 | 0,0 | 0,0 |
| 08 JAMA | 0,0 | 0,0 | 84,3 | 95,7 | 100 | 100 | 31,4 | 0,0 | 0,0 | 0,0 | 0,0 | 0,0 |
| 16 NARANJAL PAGUA | 0,0 | 0,0 | 28,8 | 85,6 | 93,7 | 93,7 | 41,4 | 30,6 | 31,5 | 18,0 | 5,4 | 5,4 |
| 15 CANAR | 0,0 | 0,0 | 38,0 | 70,9 | 78,5 | 82,3 | 36,7 | 35,4 | 32,9 | 6,3 | 0,0 | 0,0 |
| 13 GUAYAS | 0,0 | 0,0 | 26,2 | 53,9 | 66,0 | 66,2 | 32,7 | 12,7 | 8,8 | 0,6 | 0,1 | 0,1 |
| 17 JUBONES | 0,0 | 0,0 | 4,3 | 16,4 | 59,3 | 67,1 | 50,0 | 42,9 | 44,3 | 3,6 | 2,9 | 2,9 |
| 11 JIPIJAPA | 0,0 | 0,0 | 51,1 | 59,1 | 75,0 | 63,6 | 62,5 | 1,1 | 0,0 | 0,0 | 0,0 | 0,0 |
| 10 PORTOVIEJO | 0,0 | 0,0 | 18,8 | 80,0 | 97,6 | 100 | 44,7 | 0,0 | 0,0 | 0,0 | 0,0 | 0,0 |
| 12 ZAPOTAL | 0,0 | 0,0 | 99,5 | 100 | 100 | 59,0 | 57,6 | 11,9 | 1,4 | 0,0 | 0,0 | 0,0 |
| 23 ISLA PUNA | 0,0 | 0,0 | 90,6 | 100 | 100 | 100 | 100 | 31,3 | 18,8 | 3,1 | 0,0 | 0,0 |
| 22 CHIRA | 0,0 | 0,4 | 49,1 | 63,7 | 67,7 | 64,6 | 25,7 | 26,1 | 21,7 | 8,4 | 8,4 | 8,4 |
| 21 PUYANGO | 0,0 | 0,0 | 56,7 | 95,0 | 100 | 100 | 55,0 | 55,0 | 55,0 | 16,7 | 16,7 | 16,7 |
| 18 SANTA ROSA | 0,0 | 0,0 | 71,4 | 97,1 | 100 | 100 | 74,3 | 71,4 | 71,4 | 42,9 | 42,9 | 42,9 |
| 05 VERDE | 0,0 | 0,0 | 0,0 | 0,0 | 0,0 | 0,0 | 0,0 | 0,0 | 0,0 | 0,0 | 0,0 | 0,0 |
| 04 CAYAPAS | 0,0 | 0,0 | 0,0 | 0,0 | 0,0 | 0,0 | 0,0 | 0,0 | 0,0 | 0,0 | 0,0 | 0,0 |
| 03 MATAJE | 0,0 | 0,0 | 0,0 | 0,0 | 0,0 | 0,0 | 0,0 | 0,0 | 0,0 | 0,0 | 0,0 | 0,0 |
| 06 ESMERALDAS | 0,0 | 0,0 | 1,0 | 4,6 | 12,7 | 18,2 | 0,1 | 0,1 | 0,0 | 0,0 | 0,0 | 0,0 |
| 07 MUISNE | 0,0 | 0,0 | 11,3 | 32,0 | 53,6 | 53,6 | 0,0 | 0,0 | 0,0 | 0,0 | 0,0 | 0,0 |
| 25 SM PUTUMAYO | 0,0 | 0,0 | 0,0 | 0,0 | 0,0 | 0,6 | 0,6 | 2,3 | 5,7 | 0,6 | 2,8 | 1,1 |
| 02 MIRA | 0,0 | 0,0 | 0,0 | 0,0 | 0,0 | 0,5 | 0,0 | 0,0 | 0,0 | 0,0 | 0,0 | 0,0 |
| 27 CUNAMBO | 0,0 | 0,0 | 0,0 | 0,0 | 0,0 | 0,0 | 0,0 | 0,0 | 0,0 | 0,0 | 0,0 | 0,0 |
| 01 CARCHI | 0,0 | 0,0 | 0,0 | 0,0 | 0,0 | 0,0 | 0,0 | 0,0 | 0,0 | 0,0 | 0,0 | 0,0 |
| 30 SANTIAGO | 0,0 | 0,0 | 1,1 | 3,0 | 6,3 | 15,6 | 3,8 | 3,5 | 3,5 | 0,0 | 0,0 | 0,0 |
| 29 MORONA | 0,0 | 0,0 | 0,0 | 1,9 | 4,7 | 8,9 | 0,0 | 0,0 | 0,0 | 0,0 | 0,0 | 0,0 |
| 28 PASTAZA | 0,0 | 0,0 | 2,5 | 3,2 | 4,9 | 12,4 | 3,5 | 3,5 | 3,3 | 0,0 | 0,0 | 0,0 |
| 26 NAPO | 0,0 | 0,0 | 0,1 | 0,0 | 0,0 | 2,4 | 0,0 | 0,2 | 0,5 | 0,0 | 0,3 | 0,1 |
| 31 CHINCHIPE | 0,0 | 0,0 | 0,0 | 0,0 | 0,0 | 0,0 | 0,0 | 0,0 | 0,0 | 0,0 | 0,0 | 0,0 |
| 19 ARENILLAS | 0,0 | 0,0 | 95,5 | 100 | 100 | 100 | 68,2 | 68,2 | 68,2 | 63,6 | 63,6 | 63,6 |

**LEGEND (a)**

| Value | Category |
|---|---|
| ≤ -1.50 | SEVERELY DRY |
| -1.49 – -1.00 | MODERATELY DRY |
| -0.99 – 0.99 | NEAR NORMAL |
| 1.00 – 1.49 | MODERATELY HUMID |
| 1.50 – 1.99 | SEVERELY HUMID |
| ≥ 2.00 | EXTREMELY HUMID |

**LEGEND (b)**

| Value |
|---|
| 0.0% |
| 0.1 – 19.9% |
| 20.0 – 39.9% |
| 40.0 – 59.9% |
| 60.0 – 99.9% |
| 100% |