# Peer review of "Effect of extreme El Niño events on the precipitations of Ecuador"

_EGUsphere, 2022_

## Author Response (AR1)

**Responses to comments from Referee 1**

**1) Normalized precipitation indices such as the SPDI are particularly useful for drought monitoring as they measure the precipitation relative to what is expected climatologically for a specific location and season. However, it is not adequate for addressing the temporal variability of hazards associated with extreme precipitation. It is not the same to have an SPDI >2 during the rainy season, which could imply pluvial and fluvial flooding and debris flows, and during the dry season, in which the precipitation is unlikely to represent a similar hazard. It is important that the authors include the analysis of indices specific for extreme precipitation events (e.g. http://etccdi.pacificclimate.org/list_27_indices.shtml)**

*Response. We do agree with this important observation from Referee 1. We have generated important previous experience regarding the applicability of the SPDI in both, the extreme wet or dry precipitation conditions (eg. Thielen et al. 2020, Thielen et al. 2021, among other works, which are cited in the original as well as in the corrected version of the manuscript). From our own experience, we realized that SPDI could be an appropriate analysis tool since El Niño generates in this region, rather than several isolated events, a prolonged and continuous extremely humid (SPDI >2) pulse encompassing, firstly the wet seasonal and, later in a lesser degree, the dry season. Precipitations generated from a mega-Niño event are expected to occur in these conditions. SPDI, as in SPI, is design to analyze accumulative processes (ie. dry or wet pulses) rather than the effects of isolated and/or short lasting extreme precipitation events. As shown by Thielen et al. 2020 and Thielen et al. 2021, as well as the present study, main hazards and affectations from the occurrence an extreme El Niño event, occur during this prolonged wet pulse, and while SPDI >2. We agree with Referee 1 that the analysis of precipitation data on a daily basis, through indices such as Rx1day, R10mm, R20mm, etc., most certainly would provide information regarding the dynamics of any fast occurring extreme events capable of generating flash floods and debris flows. The objective of the present study was to identify, from a monthly basis analysis, the differences in the spatial-temporal dynamics of precipitation anomalies from El Niño events. We consider our research a key contribution as to identify which specific time and geographical regions along the Ecuador have the highest risk for the occurrence of extreme precipitations capable of generating flash floods and debris flows, and in which of these areas and time spams should a higher (time) resolution analysis (eg. daily basis precipitation indices) are most needed. This specific issue is precisely what we have identified as the next necessary upcoming phase of our research dealing with study at different scales of analysis the complexity of precipitation responses due to the effects of prolonged wet pulses of the different mega-Niño events.*

**2) It is necessary that that the authors include a validation of the CHIRPS satellite product using rain gauge data, at least for a few representative sites, particularly in arid regions where it might be less reliable, and for extreme events on a daily scale (e.g. https://doi.org/10.1016/j.pce.2022.103184).**

*Response. We completely agree with this important recommendation from Referee 1. Thus, the following text was added in the section 2.2 Data, citing relevant evidence regarding validation of the CHIRPS satellite product using rain gauge data at different geographic regions and conditions (please, see highlighted text in lines 16 to 18 of page 4 of the new version of the manuscript):*

> *"Precipitation layers derived from interpolations of data from climate gauge networks has proven to have some limitations (Deblauwe et al., 2016). CHIRPS provides reliable precipitation observations with high accuracy and is particularly suitable for areas with few rainfall gauges (Paredes-Trejo et al., 2016; Beck et al., 2017), especially over montane (López-Bermeo et al. 2022) or arid regions (Paredes-Trejo et al. 2017; Ramoni-Perazzi et al. 2021) where extreme events may be rather common."*

*Now, these modifications in the text implies four new citations. Thus, they have been accordingly included (as highlighted text) in the REFERENCES section of the new version of the manuscript. The new added citations are:*

Deblauwe, V., Droissart, V., Bose, R., Sonké, B., Blach-Overgaard, A., Svenning, J.-C., Wieringa, J. J., Ramesh, B. R., Stévart, T., and Couvreur, T. L. P.: Remotely sensed temperature and precipitation data improve species distribution modelling in the tropics. Glob. Ecol. Biogeogr., 25, 443–454. https://doi.org/10.1111/geb.12426, 2016.

López-Bermeo, C., Montoya, R. D., Caro-Lopera, F. J., and Díaz-García, J. A.: Validation of the accuracy of the CHIRPS precipitation dataset at representing climate variability in a tropical mountainous region of South America, Phys. Chem. Earth, 127, 10.1016/j.pce.2022.103184, 2022.

Paredes-Trejo, F. J., Alves-Barbosa, H., and Lakshmi-Kumar, T. V.: Validating CHIRPS-based satellite precipitation estimates in Northeast Brazil, Journal of Arid Environments, 139, 26-40, 2017.

Ramoni-Perazzi, P., Passamani, M., Thielen, D. R., Padovani, C., and Arizapana, M. A.: BrazilClim: The overcoming of limitations of preexisting bioclimate data. Int. J. Clim., 42, doi: 10.1002/joc.7325, 2021.

*Regarding extreme events on a daily scale, it is projected that for next necessary step in this research is to analyze the dynamics of daily precipitations. Several authors state that CHIRPS may be also considered a reliable satellite-based rainfall data source for many geographical regions (eg. Valdés-Pineda et al. 2016; Baez-Villanueva et al. 2018).*

**3) The study by Kiefer and Karamperidou (2019; https://doi.org/10.1029/2018PA003423) is very relevant and the authors should compare the results to theirs in the manuscript.**

*Response. We completely agree with the recommendation of Referee 1 about comparing our results to those from Kiefer and Karamperidou (2019). On this regard, in the Discussion section, references have been made regarding the similarities in the results between both studies specific to the effects of different ENSO flavors on precipitation dynamics, as well as to the altimetric response of precipitation anomalies to EP and COA events. For instance, from line 3 to 6 of page 14 of the new version of the manuscript, the following text was added:*

*"According to Kiefer and Karamperidou (2019), during EP and COA warm events, the coastal region is prone to extreme precipitation associated with convective bursts originating from the Pacific, while during a warm CP El Niño, as well as during a cold La Niña, moisture originates from the Atlantic and may reach the area as broader-scale less-intense precipitation."*

*An additionally reference to Kiefer and Karamperidou (2019) was made in the Discussion section (see line 31 of page 14 of the new version of the manuscript), regarding comparisons between their results with those of present research.*

*The citation Kiefer and Karamperidou (2019) was properly added (as highlighted text) to the REFERENCES section as:*

*Kiefer, J., and Karamperidou, C.: High-resolution modeling of ENSO-induced precipitation in the tropical Andes: Implications for proxy interpretation. Paleoceanography and Paleoclimatology, 34, 217–236, doi:10.1029/2018PA003423, 2019.*

**Responses to comments from Referee 2**

**1. Why you use SPDI index to monitor precipitation spatial-temporal dynamics? It seems that this index is more appropriate for drought monitoring instead of precipitation monitoring. In the introduction part and material part, there is no review about the applicability of SPDI for extreme precipitation monitoring.**

*Response*. The SPDI is very similar to the SPI (Standardized Precipitation Index). Although it has been frequently used for analyzing drought evolution, as in SPI, its application is not limited to dry events. It is a monthly rainfall index that is based on the calculation of accumulated monthly rainfall anomalies. The index is especially sensitive to any prolonged (wet or dry) precipitation event. Long lasting El Niños events are very well identified and characterized, spatial and temporarily, by this simple to use precipitation index. SPDI is simple enough to be able to be developed routinely and in real time in wide spaces and for numerous observation stations, which makes it a useful for implementation in the monitoring and forecasting of unusually dry and wet conditions at different time and space scales of analysis. We have generated important experience regarding the applicability of the SPDI in both, the extreme wet or dry precipitation conditions (eg. Thielen et al. 2020 and Thielen et al. 2021, both works cited in the original version of the manuscript). In the new version of the manuscript, here provided, additional arguments about the use of the SPDI have been included in the Material & Methods section (see lines 7-12, page 5). More specifically, the inserted text states:

> "One of the main advantages of using the SPDI is that it reflects precisely the beginning and end of each extreme precipitation event, as well as continuous information about its duration and intensity (Sanchez-Toribio et al. 2010). This ability makes it particularly suitable for characterizing long-lasting extreme events such as ENSO. Differently than the SPI, the SPDI does not require its application at multiple time scales to reflect the different durations of extreme events (Peña-Gallardo, 2016). The SPDI curves sometimes explain wet and dry periods not indicated by the SPI curves (Mega and Medjerab, 2021)."

*Now, these modifications in the text implies three new citations. Thus, they have been accordingly included (as highlighted text) in the REFERENCES section of the new version of the manuscript. The new added citations are:*

> Mega, N., and Medjerab, A.: Statistical comparison between the standardized precipitation index and the standardized precipitation drought index. Model. Earth Syst. Environ., 7, 373–388, doi:10.1007/s40808-021-01098-4, 2021.
>
> Peña-Gallardo, M., Gámiz-Fortis, S. R., Castro-Díez, Y., and Esteban-Parra, M. J.: Análisis comparativo de índices de sequía en Andalucía para el periodo 1901-2012, Cuad. de Investig. Geogr., 42, 67-88, doi:10.18172/cig.2946, 2016.
>
> Sanchez-Toribio, M. I., Garcia-Marin, R., Conesa-Garcia, C., and Lopez-Bermudez, F.: Evaporative demand and water requirements of the principal crops of the Guadalentín valley (SE Spain) in drought periods, Span. J. Agric. Res., 8, S66-S75, doi:10.5424/sjar/201008S2-1349, 2010.

**2. Following above comment, as shown in Figure 2. (Ia, Ib, IIa and IIb), even though most precipitation extremes occur in the first half of the second year of the event, the SPDI index still indicates very humid in the second half year especially in the 82/83 and 97/98 El Niño events, under the situation that the precipitation is almost zero. Thus, I further suspect the applicability of SPDI for extreme precipitation monitoring in this study.**

*Response*. As in SPI, the SPDI is a monthly rainfall index that is based on the calculation of accumulated monthly rainfall anomalies. At this time scale of analysis SPDI tends to gravitate toward zero unless a distinctive wet or dry trend is taking place. SPDI as in SPI-12 is design not to analyze the effects of isolated extreme precipitation events, but rather the accumulative effect of anomalously wet pulse(s) to which a high concomitant SPDI value can be associated with flooding and other hazards in the event of any additional precipitation. Additionally and as stated in the aforementioned inserted text, in the present research, the use of the SPDI over other traditional indexes has several advantages. First, it reflects precisely the beginning and end of each extreme precipitation event. Second, the unique ability of generating continuous information about its duration and intensity of an extreme event makes the SPDI a particularly suitable precipitation index for characterizing long-lasting extreme events such as ENSO. Third, and differently than the SPI, the SPDI does not require its application at multiple time

*scales to reflect the different durations of extreme events. Fourth, the SPDI curves sometimes explain wet and dry periods not indicated by the SPI curves.*

**3. Labels in Figure 2 and Figures should be clearer.**

*Response. Following Referee 2 recommendation, labels and characters in Figures 1, 2 and 3 have been modified and much clearer now. Resulting figures have been included in the new version of the manuscript, here provided (please, see pages 25, 26 and 27, respectively).*

**4.Obviously SPDI index is a very important part in this study, but in the abstract, there is no introduction about SPDI. Please add the information of SPDI in the abstract.**

*Response. Although we do agree with Referee 2 about the strategic value of including additional and specific information about SPDI in the Abstract, this section has already reached the maximum allowed number of words (ie. 200 words). Any case, the present study is about the analysis of precipitation anomalies generated by extreme El Niño events, and not about the analysis tool itself. In the current form, in the Abstract, the results from such anomaly analysis are sufficiently addressed. As mentioned in our response to comment 1, in the Material and Methods section of the new version of the manuscript, additional information has been included regarding different medullar aspects of the SPDI, such as its applicability and performance regarding other important precipitation anomaly index such as the SPI.*

**5.In session 2.4, why not re-sample SPDI estimation at 30 m resolution instead of resampling DEM at 0.05°, by which the altitudinal dynamics estimations would be more correct.**

*Response. When working with rasterized information there is the premise that the resampling is to the coarsest resolution of the input grids. Thus, as for the present research, all resampling must be performed towards the resolution of the precipitation anomaly data (ie. CHIRPS with 0.05°), and not that of the DEM (ie. 30x30m).*

**6.Because Central Pacific El Niños, Eastern Pacific El Niño, and Coastal El Niño are mentioned many times in the paper, please make a figure to depict where are the regions of Central Pacific El Niños, Eastern Pacific El Niño, and Coastal El Niño.**

*Response: For an El Niño event to be categorized as CP, EP or COA it depends on its SSTA pattern. Along an ENSO event, the warming and presence of sea water may be temporal and spatial very dynamic. Such dynamics may vary a lot, not only among the different types of El Niño (CP, EP vs. COA), but also among El Niño events of the same type (eg. EP-EN82/83 vs. EP-EN97/98). These pattern or dynamics is a topic under great deal of analysis and discussion. Regarding the objectives of the present study, it is out of our reach to elaborate a map (a sequence of maps, really) depicting the specific oceanic areas where each of the mega-Niño event, considered in the present study, expressed its SSTA dynamics during its entire evolution. Any case, as stated in the text of the manuscript (line 3, page 7, of the new version), readers can relate the CP-EN events occurring mainly in the central Pacific NIÑO 3.4 region, while the EP-EN and COA-EN occurring to easterly Pacific region of NIÑO 1+2 region (Larkin and Harrison, 2005; Ashok et al., 2007; Kug et al., 2009).*

**7.In the page 9. Line 17 to 18, it is confused that the sum of precipitation is only 17%, not 100%. Please clarify it.**

*Response. If March-July (n=5) have 10% each, and the rest of the months (n=7) is around 7% each, the sum is not 17%, but: (10% x 5months) + (7% x 7months) ≈ 100%. Now, to prevent any misinterpretation the text in the original manuscript (page 9, lines 17 to 18) has been rephrased as: "The monthly precipitation from March to July is about 10%, that is 50% of annual total amount. As for the rest of the year, that is from August to February, precipitation discretely drops to around 7% per month."*

**Responses to comments from the Editor**

**1. Colored or marked text in *.pdf manuscript file is not allowed. Please provide a clean version of *pdf manuscript file (with black text).**

*Response. All colored text has been eliminated from the original *.pdf manuscript and a new clean version is here provided.*

**2. Your tables contain colored cells or/and colored values. Please note that this will not be possible in the final revised version of the paper due to HTML conversion of the paper. When revising the final version, you can use footnotes or italic/bold font. But if the color spectrum is necessary and cannot be exchanged for footnotes, bold, or italic, then please inform us via email.**

*Response. All colored cells and/or colored values have been replaced by bold font in Tables 2, 3 and 4. Please, refer to the new version of the manuscript here provided.*

**3. For the next revision, please check if your figures containing maps/aerial images require a copyright statement/image credit and add it to the figures (or captions) (https://publications.copernicus.org/for_authors/manuscript_preparation.html#mapsaerials). If these figures were entirely created by the authors, there is no need to add a copyright statement or credit. In that case it is important that you confirm this explicitly by email.**

*Response. All figures in the present manuscript were entirely created by the authors, there is no need to add a copyright statement or credit.*

---

## Author Response (AR2)

**Point-by-point reply to the comments from Referee 3**

We agree with all the observations made by Referee 3, and we feel that including them in this new version of the manuscript resulted in a significant improvement of our research. Following, is a detailed description about the modifications we performed to the manuscript as a response to each of the eight comments from Referee 3. Together with this information, we are also providing a line-by-line detailed list of all the changes performed to the new version of the manuscript, here also provided.

**1) Many more details are needed to understand how the SPDI works. Specifically, a) how long do you aggregate in time (eq. 2)? since the beginning of the timeseries? b) in page 5 line 5 you say that SPDI is similar to SPI12. Why 12? Are you accumulating (eq. 2) only for 12 months? is there something I am missing? c) line 19 same page: what does it mean from i positive to i negative? what is i? these questions make it clear that you need to describe better this index. Otherwise, the non-expert user will not understand what is happening.**

*Response. We do agree with this important observation from Referee 3 that the non-expert may need additional information as to fully understand the logic behind the SPDI index. Thus, in the new version of the manuscript, page 5 lines 25-29, we added the following text:*

> *"In short, the first stage estimates the rainfall anomaly for each of the months of the series. The second stage allows to identify and calculate the accumulated rainfall anomalies, from the first month in which there is a (positive or negative) rainfall anomaly until – as a result of the accumulations – an opposite accumulated anomaly is detected. Finally, the third phase, these accumulated anomalies were standardized by converting them into z scores. Such standardization allows obtaining universally valid and comparable values for different precipitation indexes, such as the SPI."*

**2) In page 6 lines 21 and 23 you define 2 times the same acronym.**

*Response. We agree that a second definition of the acronym SST was pointless here. Thus, only the corresponding acronym was referred in the text. Please, refer to page 6 line 28 of the new version of the manuscript*

**3) I found the reading extremely difficult due to the use of boldface long 'abbreviations' to distinguish between only 4 events. Please use shorter and more reader friendly symbols and avoid boldface. For instance, why not EP83, EP98, MIX15 and COA17? The same holds for the names of the catchments. Is it necessary to have both name and number?**

*Response. Along the entire manuscript, as well as in the figures, we used the four abbreviations suggested by the referee, that is EP83, EP98, MIX15 and COA17. As for the catchments, only the names were taken into account in the text. Likewise, such numbers were eliminated in Tables 2 to 4. We do agree that our manuscript is more reader friendly now after performing these two changes.*

**4) page 7 and following, usually the symbol for p-values is not in capitals.**

*Response. We have changed to lowercase all p-value symbols in the new version of the manuscript.*

**5) page 8 line 1. I don't follow how p=0.097 is considered significant when before you said you were looking for p<0.05. please rephrase**

*Response. In the new version of the manuscript (please, refer to page 8 lines 3-4) the sentence was rephrased as:*

*"As for the Coastal El Niño, COA17, annual precipitation for the year 2017 was not significantly different from historical mean (2072mm, p=0.097)."*

**6) please change the labels in Fig. 2 to a, b, c. having I-a II-a is confusing.**

*Response. In Fig. 2 of the new version of the manuscript, we have changed the original labels to just (a) and (b). Where, (a) refers to the mean monthly precipitation, and (b) to the SPDI. Changes were also carefully made to any text referring to the graphs of this figure.*

**7) Fig. 2: it seems there is a mistake in the map in the last panel: the title says Amazon slope but the inset shows the Pacific slope.**

*Response. Indeed, there was a mistake in the inset of Fig. 2. In the new version of the manuscript such mistake was corrected.*

**8) page 11: why are the events discussed in order that is not chronological nor of event type? what is the rationale for having 3.4.3 before 3.4.4?.**

*Response. Indeed, section 3.4.3 was wrongly placed. In the new version of the manuscript (please, refer to page 13 lines 13-19) this section was placed accordingly, between sections 3.4.2 ad 3.4.4.*

*March 17th, 2023*

**Line-by-line list of changes performed to the latest version of the manuscript**

| Page | Line | Change |
|---|---|---|
| 1 | 2 | A new coauthor, Ezequiel Zamora-Ledezma, was included. |
| 1 | 9-10 | The affiliation from the new coauthor was included. |
| 1 | 3-4, 11, 13-15, 17 | Corrections to affiliations numerations were made due to the inclusion of the new coauthor. |
| 1 | 26 | "**EP-EN 97/98**" was changed to "EP98", "**COA-EN 17**" was changed to "COA17", and "**EP-EN 82/83**" was changed to "EP83". From here after, following recommendations from Referee 3, all the acronyms **EP-EN 82/83, EP-EN 97/98, MIX-EN 15/16**, and **COA-EN 17**, were changed to EP83, EP98, MIX16 and COA17, respectively. This change in the acronyms was performed along the entire manuscript, including some of the figures. |
| 5 | 25-29 | This paragraph was inserted following Referee 3 recommendations. |
| 6 | 11 | "ie." was changed for "*i.e.*" |
| 6 | 28 | Following recommendations from Referee 3, sea surface temperatures was referred here only its acronym (SST). |
| 7 | 5 | An extra parenthesis was eliminated here. |
| 7 | 12-13 | "**EP-EN 82/83**" was changed to "EP83", "**EP-EN 97/98**" was changed to "EP98", and "**MIX-EN 15/16**" was changed to "MIX16".
Also, the sentence was rephrased from "**EP-EN 82/83**, for El Niño 1982/83; **EP-EN 97/98**, for El Niño 1997/98; and **MIX-EN 15/16**, for El Niño 2015/16" to "EP83, for the EP El Niño event from 1982/83; EP98, for the EP El Niño from 1997/98; and MIX16, for the mixed CP/EP El Niño event from 2015/16". |
| 7 | 19 | "**COA-EN 17**" was changed to "COA17" |
| 7 | 26 | "Fig. 2-Ia" changed to "Fig. 2a". From here on, following recommendations from Referee 3, the labels changed from 2-Ia, 2-IIa and 2-IIIa, to simply 2a. On the other side, the labels 2-Ib, 2-IIb, and 2-IIIb, changed to 2b. Changes were made accordingly to labels from Fig. 2. |
| 7 | 27 | "P" changed to "*p*". From here on, following recommendations from Referee 3, all p-value symbols were changed to lowercase. |
| 7 | 29 | "**EP-EN 82/83**" changed to "EP83". "**EP-EN 97/98**" changed to "EP98" |
| 7 | 29 | "ie." was changed for "*i.e.*" |
| 7 | 30 | "P" changed to "*p*". |
| 7 | 31 | "**EP-EN 82/83**" changed to "EP83". "**EP-EN 97/98**" changed to "EP98" |
| 7 | 31 | "ie." was changed for "i.e." |
| 8 | 1 | "was" changed for "were" |
| 8 | 1 | "**EP-EN 82/83**" changed to "EP83". "**EP-EN 97/98**" changed to "EP98" |
| 8 | 1 | "P" changed to "p" |
| 8 | 2 | "**MIX-EN 15/16**" was changed for "MIX16" |
| 8 | 3 | "P" changed to "p" |
| 8 | 3 | "**COA-EN 17**" was changed to "COA17" |

| 8 | 3-4 | Following recommendations from Referee 3, the sentence "annual precipitation for the year 2017 tended to be significantly different from historical mean (2072mm, P=0.097)" was rephrased as "annual precipitation for the year 2017 was not significantly different from historical mean (2072mm, p=0.097)" |
|---|---|---|
| 8 | 4 | "P" changed to "p" |
| 8 | 6 | "Fig. 2-Ib" changed to "Fig. 2b" |
| 8 | 7 | "Fig. 2-Ia" changed to "Fig. 2a" |
| 8 | 9 | "**EP-EN 82/83**" changed to "EP83". "**EP-EN 97/98**" changed to "EP98" |
| 8 | 10 | "P" changed to "p" |
| 8 | 10 | "**EP-EN 82/83**" changed to "EP83" |
| 8 | 11 | "**EP-EN 82/83**" changed to "EP83" |
| 8 | 13 | "Fig. 3-I" changed to "Fig. 3a" |
| 8 | 15 | "**EP-EN 97/98**" changed to "EP98" |
| 8 | 17 | "Fig. 2-Ib" changed to "Fig. 2b" |
| 8 | 17 | "**EP-EN 97/98**" changed to "EP98" |
| 8 | 18 | "**EP-EN 82/83**" changed to "EP83" |
| 8 | 19 | "**EP-EN 97/98**" changed to "EP98" |
| 8 | 19-20 | "Fig. 3-II" changed to "Fig. 3b" |
| 8 | 22 | "**MIX-EN 15/16**" was changed for "MIX16" |
| 8 | 22 | "ie." was changed for "i.e." |
| 8 | 23 | "P" changed to "p" |
| 8 | 23 | "**EP-EN 82/83**" changed to "EP83". "**EP-EN 97/98**" changed to "EP98" |
| 8 | 24 | "P" changed to "p" |
| 8 | 25 | "Fig. 2-Ib" changed to "Fig. 2b" |
| 8 | 26 | "**COA-EN 17**" was changed to "COA17" |
| 8 | 27 | "Fig. 2-Ib" changed to "Fig. 2b" |
| 8 | 28 | "**EP-EN 82/83**" changed to "EP83". "**EP-EN 97/98**" changed to "EP98" |
| 8 | 28 | "P" changed to "p" |
| 8 | 29 | "Fig. 3-III" changed to "Fig. 3c" |
| 8 | 30 | "**COA-EN 17**" was changed to "COA17". "**EP-EN 82/83**" changed to "EP83". "**EP-EN 97/98**" changed to "EP98" |
| 9 | 4 | "Fig. 2-IIa" was changed to "Fig. 2a" |
| 9 | 6 | "P" changed to "p" |
| 9 | 7 | "**EP-EN 82/83**" changed to "EP83". "**EP-EN 97/98**" changed to "EP98" |
| 9 | 9 | "P" changed to "p" |
| 9 | 9 | "**EP-EN 82/83**" changed to "EP83". "**EP-EN 97/98**" changed to "EP98" |
| 9 | 10 | "P" changed to "p" |
| 9 | 10 | "**MIX-EN 15/16**" was changed for "MIX16" |
| 9 | 11 | "P" changed to "p" |
| 9 | 12 | "P" changed to "p" |
| 9 | 12 | "**COA-EN 17**" was changed to "COA17" |
| 9 | 13 | "P" changed to "p" |
| 9 | 16 | "**EP-EN 82/83**" changed to "EP83". "**EP-EN 97/98**" changed to "EP98" |
| 9 | 17 | "Fig. 2-IIb" changed to "Fig. 2b" |
| 9 | 17 | "Fig. 3-I and II " changed to "Fig. 3a and b" |
| 9 | 18 | "P" changed to "p" |

| 9 | 18 | "**EP-EN 82/83**" changed to "EP83". "**EP-EN 97/98**" changed to "EP98" |
|---|---|---|
| 9 | 24 | "**EP-EN 97/98**" changed to "EP98" |
| 9 | 25 | "**MIX-EN 15/16**" was changed for "MIX16" |
| 9 | 27 | "Fig. 2-IIb" changed to "Fig. 2b" |
| 9 | 27 | "Fig. 3-III" changed to "Fig. 3c" |
| 9 | 28 | "P" changed to "p" |
| 9 | 28 | "Fig. 2-Ib and IIb" changed to "Fig. 2b" |
| 9 | 29 | "P" changed to "p" |
| 10 | 5 | "Fig. 2-IIIa" changed to "Fig. 2a" |
| 10 | 7 | "**EP-EN 97/98**" changed to "EP98" |
| 10 | 8 | "**MIX-EN 15/16**" was changed for "MIX16" |
| 10 | 8 | "P" changed to "p" |
| 10 | 9 | "**EP-EN 82/83**, **EP-EN 97/98**, **MIX-EN 15/16** or **COA-EN 17**" changed to "EP83, EP98, MIX16, or COA17" |
| 10 | 10 | "P" changed to "p" |
| 10 | 10 | "**MIX-EN 15/16**" was changed for "MIX16" |
| 10 | 10 | "**EP-EN 82/83**, **EP-EN 97/98** and **COA-EN 17**" changed to "EP83, EP98, and COA17" |
| 10 | 14 | "Fig. 2-IIIb" changed to "Fig. 2b" |
| 10 | 14 | "**EP-EN 97/98**, **MIX-EN 15/16**, and **COA-EN 17**" changed to "EP98, MIX16, and COA17" |
| 10 | 15 | "P" changed to "p" |
| 10 | 15 | "**EP-EN 82/83**" changed to "EP83" |
| 10 | 16 | "**EP-EN 82/83**" changed to "EP83" |
| 10 | 16 | "**EP-EN 97/98**" changed to "EP98" |
| 10 | 18 | "P" changed to "p" |
| 10 | 18 | "**EP-EN 82/83**" changed to "EP83" |
| 10 | 18 | "**EP-EN 97/98**, **MIX-EN 15/16**, and **COA-EN 17** " changed to "EP98, MIX16, and COA17" |
| 10 | 18 | "P" changed to "p" |
| 10 | 18-19 | "Fig. 2-IIIb" changed to "Fig. 2b" |
| 10 | 20 | "**EP-EN 82/83**" changed to "EP83" |
| 10 | 20 | "**EP-EN 97/98**" changed to "EP98" |
| 10 | 20 | "**COA-EN 17**" was changed to "COA17" |
| 10 | 20 | "Fig. 3-I, II and III" changed to "Fig. 3a, b and c" |
| 10 | 21 | "**MIX-EN 15/16**" was changed for "MIX16" |
| 10 | 21 | "**EP-EN 97/98**" changed to "EP98" |
| 10 | 22 | "**COA-EN 17**" was changed to "COA17" |
| 10 | 23 | "**EP-EN 82/83**" changed to "EP83" |
| 10 | 25 | "**EP-EN 82/83**" changed to "EP83" |
| 10 | 25 | "**EP-EN 97/98**" changed to "EP98" |
| 10 | 26 | "**COA-EN 17**" was changed to "COA17" |
| 11 | 2 | "3.4.1 EP-EN 82/83" changed to "3.4.1 Easter Pacific El Niño event EP83" |
| 11 | 4-5 | "16-NARANJAL PAGUA, 23-ISLA PUNA, 17-JUBONES, 14-TAURA, 13-GUAYAS, 15-CANAR, 12-ZAPOTAL, 21-PUYANGO, 18-SANTA ROSA, 19-ARENILLAS, and 20-ZARUMILLA" changed to "NARANJAL PAGUA, ISLA PUNA, JUBONES, TAURA, GUAYAS, CANAR, ZAPOTAL, PUYANGO, SANTA ROSA, ARENILLAS, and ZARUMILLA". Following Referee 3 instructions, the numbers of the catchments were eliminates along the text of the manuscript, as well as in Tables 2, 3 and 4. |
| 11 | 6-7 | "Table 2-a" changed to "Table 2a" |

| 11 | 7 | "Table 2-b" changed to "Table 2b" |
|---|---|---|
| 11 | 8 | "11-JIPIJAPA, 09-CHONE, 10-PORTOVIEJO, 07-MUISNE, and 08-JAMA" changed to "JIPIJAPA, CHONE, PORTOVIEJO, MUISNE, and JAMA" |
| 11 | 9 | "Table 2-a" changed to "Table 2a" |
| 11 | 10 | "P" changed to "p" |
| 11 | 11 | "Table 2-b" changed to "Table 2b" |
| 11 | 12 | **"EP-EN 82/83"** changed to "EP83" |
| 11 | 13 | "P" changed to "p" |
| 11 | 15 | "29-MORONA, 28-PASTAZA, and 30-SANTIAGO" changed to "MORONA, PASTAZA, and SANTIAGO" |
| 11 | 16 | "Table 2-a" changed to "Table 2a" |
| 11 | 19 | "31-CHINCHIPE, 26-NAPO, 27-CUNAMBO, and 25 SM PUTUMAYO" changed to "CHINCHIPE, NAPO, CUNAMBO, and SM PUTUMAYO" |
| 11 | 19-20 | "01-CARCHI, 02-MIRA, 05-VERDE, 03-MATAJE, 04-CAYAPA and 06-ESMERALDAS" changed to "CARCHI, MIRA, VERDE, MATAJE, CAYAPA and ESMERALDAS" |
| 11 | 20 | "22-CHIRA" changed to "CHIRA" |
| 11 | 21 | "Table 2-a" changed to "Table 2a" |
| 11 | 23 | "Table 2-b" changed to "Table 2b" |
| 11 | 24 | "3.4.2 EP-EN 97/98" changed to "3.4.2 Easter Pacific El Niño event EP98" |
| 11 | 25 | **"EP-EN 97/98"** changed to "EP98" |
| 11 | 27-28 | "14-JUBONES, 06-ESMERALDAS, 21-PUYANGO, 08-JAMA, 07-MUISNE, 13-GUAYAS, 16-NARANJAL PAGUA, 14-TAURA, 15-CANAR, 12-ZAPOTAL, 19-ARENILLAS, 18-SANTA ROSA, 20-ZARUMILLA, and 23-ISLA PUNA" changed to "JUBONES, ESMERALDAS, PUYANGO, JAMA, MUISNE, GUAYAS, NARANJAL PAGUA, TAURA, CANAR, ZAPOTAL, ARENILLAS, SANTA ROSA, ZARUMILLA, and ISLA PUNA" |
| 11 | 30 | "Table 3-a" changed to "Table 3a" |
| 11 | 31 | "06-ESMERALDAS" changed to "ESMERALDAS" |
| 12 | 1 | "Table 3-b" changed to "Table 3b" |
| 12 | 2 | "10-PORTOVIEJO, 09-CHONE and 11-JIPIJAPA" changed to "PORTOVIEJO, CHONE and JIPIJAPA" |
| 12 | 5 | "Table 3-a" changed to "Table 3a" |
| 12 | 6 | "Table 3-b" changed to "Table 3b" |
| 12 | 7 | **"EP-EN 97/98"** changed to "EP98" |
| 12 | 9 | "03-MATAJE, 02-MIRA, 29-MORONA, 28-PASTAZA, 27-CUNAMBO, 30-SANTIAGO, 04-CAYAPA, 05-VERDE, and 22-CHIRA" changed to "MATAJE, MIRA, MORONA, PASTAZA, CUNAMBO, SANTIAGO, CAYAPA, VERDE, and CHIRA" |
| 12 | 10 | "Table 3-a" changed to "Table 3a" |
| 12 | 11 | "26-NAPO, 25-SM PUTUMAYO, 31-CHINCHIPE, and 01-CARCHI" changed to "NAPO, SM PUTUMAYO, CHINCHIPE, and CARCHI" |
| 12 | 12 | "Table 3-a" changed to "Table 3a" |
| 12 | 13 | "3.4.3 MIX-EN 15/16" changed to "3.4.3 Mixed El Niño event MIX16"
 Also, this entire section was moved from between sections 3.4.1 and 3.4.2 to its current position, between sections 3.4.2 and 3.4.4. |
| 12 | 14 | **"MIX-EN 15/16"** was changed for "MIX16" |
| 12 | 15-16 | "08-JAMA, 09-CHONE, 19-ARENILLAS, 21-PUYANGO, and 20-ZARUMILLA" changed to "JAMA, CHONE, ARENILLAS, PUYANGO, and ZARUMILLA" |
| 12 | 18 | "31-CHINCHIPE" changed to "CHINCHIPE" |

| 12 | 20 | "3.4.4 COA-EN 17" changed to "3.4.4 Coastal El Niño event COA17" |
|---|---|---|
| 12 | 21-23 | "20-ZARUMILLA, 14-TAURA, 09-CHONE, 08-JAMA, 16-NARANJAL PAGUA, 15-CANAR, 13-GUAYAS, 17-JUBONES, 11-JIPIJAPA, 10-PORTOVIEJO, 12-ZAPOTAL, 23-ISLA PUNA, 22-CHIRA, 21-PUYANGO, and 18-SANTA ROSA" changed to "ZARUMILLA, TAURA, CHONE, JAMA, NARANJAL PAGUA, CANAR, GUAYAS, JUBONES, JIPIJAPA, PORTOVIEJO, ZAPOTAL, ISLA PUNA, CHIRA, PUYANGO, and SANTA ROSA" |
| 12 | 23 | "Table 4-a" changed to "Table 4a" |
| 12 | 25 | "Table 4-a" changed to "Table 4a" |
| 12 | 26 | "18-SANTA ROSA" changed to "SANTA ROSA" |
| 12 | 27 | "**COA-EN 17**" was changed to "COA17" |
| 12 | 28-29 | "05-VERDE, 04-CAYAPAS, 03-MATAJE, 06-ESMERALDAS, 07-MUISNE, 25-SM PUTUMAYO, 02-MIRA, 27-CUNAMBO, 01-CARCHI, and 30-SANTIAGO" changed to "VERDE, CAYAPAS, MATAJE, ESMERALDAS, MUISNE, SM PUTUMAYO, MIRA, CUNAMBO, CARCHI, and SANTIAGO" |
| 12 | 29 | "29-MORONA, 28-PASTAZA, 26-NAPO, and 31-CHINCHIPE" changed to "MORONA, PASTAZA, NAPO, and CHINCHIPE" |
| 12 | 29 | "Table 4-a" changed to "Table 4a" |
| 13 | 1 | "19-ARENILLAS" changed to "ARENILLAS" |
| 13 | 3 | "Table 4-a and b" changed to "Table 4a and b" |
| 13 | 11 | "**EP-EN 82/83, EP-EN 97/98, MIX-EN 15/16,** and **COA-EN 17**" changed to "EP83, EP98, MIX16, and COA17" |
| 13 | 13 | "**EP-EN 82/83**, **EP-EN 97/98**" changed to "EP83, EP98" |
| 13 | 13 | "**COA-EN 17**" was changed to "COA17" |
| 13 | 13 | An extra parenthesis was eliminated here. |
| 13 | 17 | "**MIX-EN 15/16**" was changed for "MIX16" |
| 13 | 24 | "ie." was changed for "*i.e.*" |
| 13 | 31 | "**MIX-EN 15/16**" was changed for "MIX16" |
| 14 | 6 | "P" changed to "p" |
| 14 | 6 | "**EP-EN 82/83** and **EP-EN 97/98**" changed to "EP83 and EP98" |
| 14 | 9 | "**EP-EN 97/98**" changed to "EP98" |
| 14 | 11 | "**EP-EN 82/83** and **COA-EN 17**" changed to "EP83 and COA17" |
| 14 | 11 | "P" changed to "p" |
| 14 | 19 | "P" changed to "p" |
| 14 | 20 | "**EP-EN 97/98**" changed to "EP98" |
| 14 | 22 | "**COA-EN 17** and **EP-EN 82/83**" changed to "COA17 and EP83" |
| 15 | 6 | "**EP-EN 82/83** and **EP-EN 97/98**" changed to "EP83 and EP98" |
| 15 | 6 | "**COA-EN 17**" was changed to "COA17" |
| 15 | 10 | "02-MIRA, 06-ESMERALDAS and part of 13-GUAYAS" changed to "MIRA, ESMERALDAS and part of GUAYAS" |
| 15 | 11-12 | "13-GUAYAS, 15-CANAR, 16-NARANJAL PAGUA, and 17-JUBONES" changed to "GUAYAS, CANAR, NARANJAL PAGUA, and JUBONES" |
| 15 | 14 | "22-CHIRA" changed to "CHIRA" |
| 15 | 29 | "24-GALAPAGOS" changed to "GALAPAGOS" |
| 15 | 30-33 | "19-ARENILLAS (SPDI 2.47 and ⁻SI 1.18), 08-JAMA (2.31 and 1.19), 18-SANTA ROSA (2.31 and 1.11), 09-CHONE (2.25 and 1.23), 12-ZAPOTAL (2.23 and 1.33), 16-NARANJAL PAGUA (2.22 and 1.10), 23-ISLA PUNA (2.22 and 1.27), 11-JIPIJAPA (2.19 and 1.27), 20-ZARUMILLA (2.17 and 1.19), 21-PUYANGO (2.16 and 1.19), 10-PORTOVIEJO (2.11 and 1.29), 14-TAURA (2.09 and 1.29), and |

| | | 13-GUAYAS (1.87 and 1.15)" changed to "ARENILLAS (SPDI 2.47 and ⁻SI 1.18), JAMA (2.31 and 1.19), SANTA ROSA (2.31 and 1.11), CHONE (2.25 and 1.23), ZAPOTAL (2.23 and 1.33), NARANJAL PAGUA (2.22 and 1.10), ISLA PUNA (2.22 and 1.27), JIPIJAPA (2.19 and 1.27), ZARUMILLA (2.17 and 1.19), PUYANGO (2.16 and 1.19), PORTOVIEJO (2.11 and 1.29), TAURA (2.09 and 1.29), and GUAYAS (1.87 and 1.15)" |
|---|---|---|
| 16 | 15 | "**EP-EN 97/98**" changed to "EP98" |
| 16 | 16 | "**COA-EN 17** and **EP-EN 82/83**" changed to "COA17 and EP83" |
| 17 | 17 | The initials EZL, which refers to the new coauthor Ezequiel Zamora-Ledezma, were added |
| 17 | 19 | The initials EZL, which refers to the new coauthor Ezequiel Zamora-Ledezma, were added |
| 17 | 20 | The text "Revision and re-validation: DT, PRP, EZL, MLP" was added here |
| 17 | 26-29 | The text "This work was also supported by the mining canon (002-2019-P.CO-UNAH/FOCAM)" was eliminated and replaced by the text "This work was also supported by the Research Institute of Universidad Técnica de Manabí (UTM), Portoviejo, Ecuador and by the Programa de movilizaciones para investigación en Cambio Climatico-AMSUD 189-2020-FONDECYT. Finally, the authors wish to thank the anonymous reviewers for their recommendations which resulted in a significant improvement of the present work" |
| 24 | 6 | In captions to Figure 1 ":" was changed to "." |
| 24 | 6 | In captions to Figure 1 "km2" was changed to "km$^2$", twice |
| 24 | 10 | In captions to Figure 1 "km2" was changed to "km$^2$" |
| 25 | 1 | As mentioned before, following recommendations from Referee 3, the labels changed from "Ia", "IIa" and "IIIa", to simply "a". On the other side, the labels "Ib", "IIb", and "IIIb", changed to "b". |
| 25 | 1 | Inside each graph 2 legend, the text "EP-EN 82/83", "EP-EN 97/98", "MIX-EN 15/16" and "COA-EN 17" was changed to the according text "EP83", "EP98", "MIX16" and "COA17" |
| 25 | 3 | In captions to Figure 2 ":" was changed to "." |
| 25 | 3 | In captions to Figure 2 "(Ia, IIa, and IIIa):" was changed to "(a)." |
| 25 | 4 | In captions to Figure 2 "**EP-EN 82/83**, **EP-EN 97/98**, **MIX-EN 15/16** and **COA-EN 17**" was changed to "EP83, EP98, MIX16 and COA17" |
| 25 | 4 | In captions to Figure 2 "(Ib, IIb, and IIIb):" was changed to "(b)." |
| 26 | 1 | Inside each graph 3 "(I)", "(II)" and "(III)" were changed to "(a)", "(b)" and "(c)", correspondingly |
| 26 | 1 | Inside each graph 3 the text "EP-EN 82/83", "EP-EN 97/98" and "COA-EN 17" was changed to the according text "EP83", "EP98" and "COA17" |
| 26 | 1 | Inside the graph AMAZON SLOPE – SPDI (b) the inset corrected to that belonging to the Amazon slope |
| 26 | 2 | In captions to Figure 3 ":" was changed to "." |
| 26 | 3 | In captions to Figure 3 "**EP-EN 82/83**" changed to "EP83" |
| 26 | 3 | In captions to Figure 3 "(I)" was changed to "(a)" |
| 26 | 3 | In captions to Figure 3 "**EP-EN 97/98**" changed to "EP98" |
| 26 | 3 | In captions to Figure 3 "(II)" was changed to "(b)" |
| 26 | 3 | In captions to Figure 3 "**COA-EN 17**" changed to "COA17" |
| 26 | 3 | In captions to Figure 3 "(III)" was changed to "(c)" |
| 27 | 5 | Inside the graph 4 the text "EP-EN 82/83", "EP-EN 97/98" and "COA-EN 17" was changed to the according text "EP83", "EP98" and "COA17" |
| 27 | 6 | In captions to Figure 4 ":" was changed to "." |
| 27 | 6 | In captions to Figure 4 "**EP-EN 82/83** and **EP-EN 97/98**, and the Coastal El Niño **COA-EN 17**" changed to "EP83 and EP98, and the Coastal El Niño COA17" |
| 28 | 11 | In captions to Figure 5 ":" was changed to "." |

| 28 | 6 | In captions to Figure 5 "**EP-EN 82/83** and **EP-EN 97/98**, and the Coastal El Niño **COA-EN 17**" changed to "EP83 and EP98, and the Coastal El Niño COA17" |
|----|----|----|
| 29 | 15 | Inside the graph 4, the text in the inset was changed all to capital letters |
| 29 | 16 | In captions to Figure 6 ":" was changed to "." |
| 29 | 17 | In captions to Figure 6 "**EP-EN 82/83** and **EP-EN 97/98**, and the Coastal El Niño **COA-EN 17**" changed to "EP83 and EP98, and the Coastal El Niño COA17" |
| 31 | 25 | In captions to Table 2 "**EP-EN 82/83**" changed to "EP83" |
| 31 | 27 | In Table 2, the numeration originally assigned to each catchment was eliminated |
| 32 | 30 | In captions to Table 3 "**EP-EN 97/98**" changed to "EP98" |
| 32 | 32 | In Table 3, the numeration originally assigned to each catchment was eliminated |
| 33 | 35 | In captions to Table 4 "**COA-EN 17**" changed to "COA17" |
| 33 | 37 | In Table 4, the numeration originally assigned to each catchment was eliminated |